# Cylicins are a structural component of the sperm calyx being indispensable for male fertility in mice and human

**Simon Schneider[1,2†], Andjela Kovacevic[1†], Michelle Mayer[1‡], Ann-Kristin Dicke[3], Lena Arévalo[1], Sophie A Koser[3], Jan N Hansen[4], Samuel Young[5], Christoph Brenker[5], Sabine Kliesch[5], Dagmar Wachten[4], Gregor Kirfel[6], Timo Strünker[5], Frank Tüttelmann[3], Hubert Schorle[1]***

[1]Institute of Pathology, Department of Developmental Pathology, Medical Faculty, University of Bonn, Bonn, Germany; [2]Bonn Technology Campus, Core Facility 'Gene-Editing', Medical Faculty, University of Bonn, Bonn, Germany; [3]Institute of Reproductive Genetics, University of Münster, Münster, Germany; [4]Institute of Innate Immunity, Biophysical Imaging, Medical Faculty, University of Bonn, Bonn, Germany; [5]Centre of Reproductive Medicine and Andrology, University Hospital Münster, University of Münster, Münster, Germany; [6]Institute for Cell Biology, University of Bonn, Bonn, Germany

*For correspondence:
Schorle@uni-bonn.de

†These authors contributed
equally to this work

Present address: ‡Life and
Medical Sciences Institute,
Department for Immunology and
Environment, University of Bonn,
Bonn, Germany

Reviewing Editor: Jean-Ju
Chung, Yale University, United
States

**Abstract** Cylicins are testis-specific proteins, which are exclusively expressed during spermio-genesis. In mice and humans, two Cylicins, the gonosomal X-linked Cylicin 1 (*Cylc1/CYLC1*) and the autosomal Cylicin 2 (*Cylc2/CYLC2*) genes, have been identified. Cylicins are cytoskeletal proteins with an overall positive charge due to lysine-rich repeats. While Cylicins have been localized in the acrosomal region of round spermatids, they resemble a major component of the calyx within the perinuclear theca at the posterior part of mature sperm nuclei. However, the role of Cylicins during spermiogenesis has not yet been investigated. Here, we applied CRISPR/Cas9-mediated gene editing in zygotes to establish *Cylc1*- and *Cylc2*-deficient mouse lines as a model to study the function of these proteins. *Cylc1* deficiency resulted in male subfertility, whereas *Cylc2−/−*, *Cylc1−/yCylc2+/−*, and *Cylc1−/yCylc2−/−* males were infertile. Phenotypical characterization revealed that loss of Cylicins prevents proper calyx assembly during spermiogenesis. This results in decreased epididymal sperm counts, impaired shedding of excess cytoplasm, and severe structural malformations, ultimately resulting in impaired sperm motility. Furthermore, exome sequencing identified an infertile man with a hemizygous variant in *CYLC1* and a heterozygous variant in *CYLC2*, displaying morphological abnormalities of the sperm including the absence of the acrosome. Thus, our study highlights the relevance and importance of Cylicins for spermiogenic remodeling and male fertility in human and mouse, and provides the basis for further studies on unraveling the complex molecular interactions between perinuclear theca proteins required during spermiogenesis.

## eLife assessment

This study provides **valuable** insights into the role of two under-researched sperm-specific proteins (Cylicin 1 and Cylicin 2). The authors provide **convincing** evidence that they have an essential role in sperm head structure during spermatogenesis, and that their loss leads to subfertility or infertility, with a dose-dependent phenotype. Importantly, the authors identify infertile males with mutations in both Cylicin1 and Cylicin2. Thus, the findings from the mouse models might be applicable to understanding human male infertility with similar structural defects.

**eLife digest** Male humans, mice and other animals produce sex cells known as sperm that seek out and fertilize egg cells from females. Sperm have a very distinctive shape with a head and a long tail that enables them to swim towards an egg. At the front of the sperm's head is a pointed structure known as the acrosome that helps the sperm to burrow into an egg cell.

A structure known as the cytoskeleton is responsible for forming and maintaining the shape of acrosomes and other parts of cells. Two proteins, known as Cylicin 1 and Cylicin 2, are unique to the cytoskeleton of sperm, but their roles remain unclear.

To investigate the role of the Cylicins during spermiogenesis, Schneider, Kovacevic et al. used an approach called CRISPR/Cas9-mediated gene-editing to generate mutant mice that were unable to produce either Cylicin 1 or Cylicin 2, or both proteins. The experiments found that healthy female mice were less likely to become pregnant when they mated with mutant males that lacked Cylicin 1 compared with males that had the protein. When they did become pregnant, the females had smaller litters of babies.

Mutant male mice lacking Cylicin 2 or both Cylicin proteins (so-called "double" mutants), were infertile and mating with healthy female mice did not lead to any pregnancies. Further experiments found that the sperm of such mice had smaller heads than normal sperm, defective acrosomes, and curled tails that wrapped around the head.

Schneider, Kovacevic et al. also examined the sperm of a human patient who had inherited genetic variants in the genes encoding both Cylicin proteins. Similar to the double mutant mice, the patient was infertile, and his sperm also had defective acrosomes and curled tails.

These findings indicate that Cylicins are required to make the acrosome as sperm cells mature and help maintain the structure of the cytoskeleton of sperm. Further studies of Cylicins and other sperm proteins in mice may help us to understand some of the factors that contribute to male infertility in humans.

## Introduction

The differentiation of round spermatids into sperm during spermiogenesis is a highly organized and spatiotemporally controlled process taking place in the seminiferous epithelium of the testis. Cellular and morphological remodeling involves DNA hypercondensation, establishment of a species-specific head morphology, removal of excess cytoplasm, as well as formation of accessory structures like acrosome and flagellum. These structural changes depend on a highly efficient protein trafficking machinery and a unique sperm cytoskeleton (*Teves et al., 2020*). One essential cytoskeletal element is the perinuclear theca (PT), which surrounds the sperm nucleus, except for the caudal edge, at the implantation site of the flagellum. The PT is supposed to serve as a structural scaffold for the sperm nucleus and resembles a rigid cytosolic protein layer, which is resistant to non-ionic detergents and high salt buffer extractions (*Longo et al., 1987*; *Longo and Cook, 1991*). The PT has been subdivided into a subacrosomal and postacrosomal part based on its localization, function, composition, and developmental origin. The subacrosomal part of the PT develops early during spermiogenesis, simultaneously with the formation of the acrosome. It is supposed to emerge from acrosomal vesicles, and presents as a thin cytosolic protein layer between the inner acrosomal membrane and the nuclear envelope (*Oko and Sutovsky, 2009*; *Oko and Maravei, 1995*). The postacrosomal part of the PT, also known as the sperm calyx, originates from cytosolic proteins that are transported via the manchette and assembled during sperm head elongation (*Oko and Sutovsky, 2009*). It is located in-between the nuclear envelope and the sperm plasma membrane.

Proteomic analyses using murine and bovine PT extracts revealed 500–800 different proteins, highlighting its high molecular complexity (*Zhang et al., 2022a*; *Zhang et al., 2022b*). Apart from cytoskeletal proteins, signaling molecules and several de novo synthesized core histones were identified. They are proposed to be essential for sperm-egg interaction and chromatin remodeling of the male pronucleus at early post-fertilization stages (*Hamilton et al., 2021*; *Sutovsky et al., 2003*; *Oko and Sutovsky, 2009*; *Herrada and Wolgemuth, 1997*; *Tovich and Oko, 2003*). Many structural proteins of the PT are testis-specific and uniquely expressed in the PT, including Calicin (*Ccin*) (*Paranko et al., 1988*; *Longo et al., 1987*; *von Bülow et al., 1995*), Cylicin 1 (*Cylc1*) and Cylicin 2 (*Cylc2*) (*Longo*

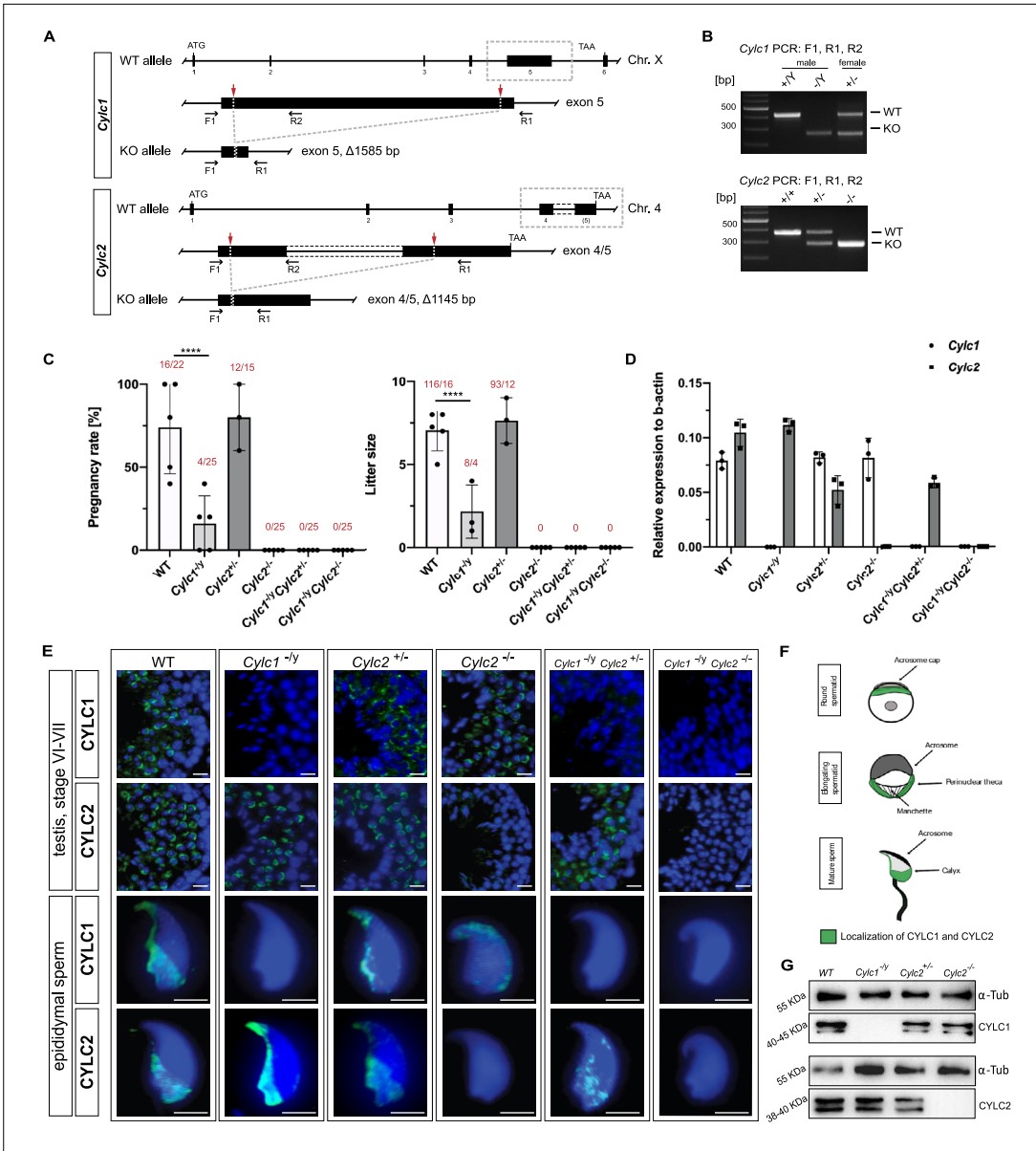

**Figure 1.** Loss of *Cylc1* or *Cylc2* results in impaired male fertility. (**A**) Schematic representation of the *Cylc1* and *Cylc2* gene structure and targeting strategy for CRISPR/Cas9-mediated generation of *Cylc1*- and *Cylc2*-deficient alleles. Targeting sites of guide RNAs are depicted by red arrows. Genotyping primer binding sites are depicted by black arrows. (**B**) Representative genotyping PCR of *Cylc1*- and *Cylc2*-deficient mice. N=3. (**C**) Fertility analysis of Cylicin-deficient mice visualized by mean litter size and pregnancy rate (%) in comparison to wildtype (WT) matings. Black dots represent mean values obtained for each male included in fertility testing. Columns represent mean values ± standard deviation (SD). Total number of offspring per total number of pregnancies as well as total number of pregnancies per total number of plugs are depicted above each bar. (**D**) Expression of *Cylc1* and *Cylc2* in testicular tissue of WT, *Cylc1⁻/ʸ*, *Cylc2⁺/⁻*, *Cylc2⁻/⁻*, *Cylc1⁻/ʸ Cylc2⁺/⁻*, and *Cylc1⁻/ʸ Cylc2⁻/⁻* mice analyzed by quantitative reverse transcription-polymerase chain reaction (qRT-PCR). Biological replicate of 3 was used. (**E**) Immunofluorescent staining of testicular tissue and cauda epididymal sperm from WT, *Cylc1⁻/ʸ*, *Cylc2⁺/⁻*, *Cylc2⁻/⁻*, *Cylc1⁻/ʸ Cylc2⁺/⁻*, and *Cylc1⁻/ʸ Cylc2⁻/⁻* males against CYLC1 and CYLC2. Cell nuclei were counterstained with DAPI. Staining was performed on three animals from each genotype. Scale bar: 5 μm. (**F**) Schematic illustration of CYLC localization during spermiogenesis. CYLC localization (green) is shown for round and elongating spermatids as well as mature sperm. (**G**) Representative immunoblot against CYLC1 and CYLC2 on cytoskeletal protein fractions from WT, *Cylc1⁻/ʸ*, *Cylc2⁺/⁻*, and *Cylc2⁻/⁻* testes. α-Tubulin was used as load control.

The online version of this article includes the following source data and figure supplement(s) for figure 1:

*Figure 1 continued on next page*

*Figure 1 continued*

**Source data 1.** PCR-genotyping of Cylicin-deficient mice.

**Source data 2.** Pregnancy rates and litter sizes of WT female mice mated to Cylicin-deficient male mice.

**Source data 3.** Cylicin1 and Cylicin2 staining of mature sperm in testis tissues.

**Source data 4.** Western-blot validation of the knockout.

**Figure supplement 1.** Amino acid sequence comparison of CYLC1 and CYLC2 in *Caenorhabditis elegans* and *Mus musculus* to *Homo sapiens*.

**Figure supplement 2.** Immunohistochemical staining against CYLC1 and CYLC2 in tissue sections of testis, brain, thymus, and spleen.

**Figure supplement 3.** Immunofluorescence staining against the acrosomal matrix marker protein SP56 (green) and CYLC1 or CYLC2 (red) in round and elongating spermatids.

**Figure supplement 4.** Immunofluorescence staining of CYLC1 and CYLC2 in elongating spermatids of wildtype (WT), *Cylc1$^{-/y}$*, *Cylc2$^{+/-}$*, *Cylc2$^{/-}$* , *Cylc1$^{-/y}$ Cylc2$^{+/-}$*, and *Cylc1$^{-/y}$ Cylc2$^{-/-}$* mice.

**Figure supplement 5.** Proteome abundances.

**Figure supplement 6.** Proteome clustering.

*et al., 1987*; *Hess et al., 1993*; *Hess et al., 1995*), actin-capping proteins CPβ3 and CPα3 (*Bülow et al., 1997*; *Hurst et al., 1998*; *Tanaka et al., 1994*), as well as actin-related proteins Arp-T1 and Arp-T2 (*Heid et al., 2002*). Despite being characterized on molecular level, their function remains largely elusive.

Both, *Ccin* and Cylicins, are highly basic proteins with a predominant localization in the calyx of mature sperm (*Hess et al., 1993*; *Hess et al., 1995*; *Paranko et al., 1988*). In most species, two Cylicin genes, *Cylc1* and *Cylc2*, have been identified (*Figure 1—figure supplement 1*). They are characterized by repetitive lysine-lysine-aspartic acid (KKD) and lysine-lysine-glutamic acid (KKE) peptide motifs, resulting in an isoelectric point (IEP) >pH 10 (*Hess et al., 1993*; *Hess et al., 1995*). Repeating units of up to 41 amino acids in the central part of the molecules stand out by a predicted tendency to form individual short α-helices (*Hess et al., 1993*). Mammalian Cylicins exhibit similar protein and domain characteristics, but CYLC2 has a much shorter amino-terminal portion than CYLC1 (*Figure 1—figure supplement 1*). While the *CYLC2/Cylc2* gene is encoded on autosomes, the human and murine *CYLC1/Cylc1* gene is encoded on the X-chromosome, resulting in hemizygosity in males. Further, in bovine, Cylc2 serves as actin-binding protein of the sperm perinuclear cytoskeleton (*Rousseaux-Prévost et al., 2003*). Cylicins seem to be cytoskeletal regulators and required for proper sperm head architecture.

In this study, we report the CRISPR/Cas9-mediated generation and characterization of *Cylc1-*, *Cylc2-*, and *Cylc1/2*-deficient mice, demonstrating that Cylicins are indispensable for male fertility and play a key function in the formation of the sperm calyx. We show that Cylicins are required for the maintenance of the PT integrity during spermiogenesis and in mature sperm. Their deficiency results in morphological defects of the sperm head, acrosome, and midpiece. Our analyses revealed that *Cylicin* genes are evolutionary conserved in rodents and primates. Furthermore, we identified Cylicin variants in an infertile man, demonstrating the conserved role of Cylicins in regulating male infertility.

## Results

### Cylicins are indispensable for male fertility in mice

Cylicins were first discovered and characterized in the early 1990s; due to their subcellular localization a role in sperm head architecture was postulated (*Longo et al., 1987*; *Hess et al., 1993*; *Hess et al., 1995*). To address the question of the role of Cylicins during spermiogenesis, we used CRISPR/Cas9-mediated gene editing to generate *Cylc1-* and *Cylc2-*deficient mouse models. First, for *Cylc1*, a pair of sgRNAs targeting exon 5 was designed, and a mouse line with a frameshift inducing deletion of 1.585 kb, accounting for 85% of the *Cylc1* coding sequence, was established (*Figure 1A*). Next, two guide RNAs targeting exon 4/5 of the *Cylc2* gene were applied to establish a *Cylc2-*deficient mouse line with a 1.145 kb frameshift inducing deletion (*Figure 1A*). In both lines, the majority of predicted

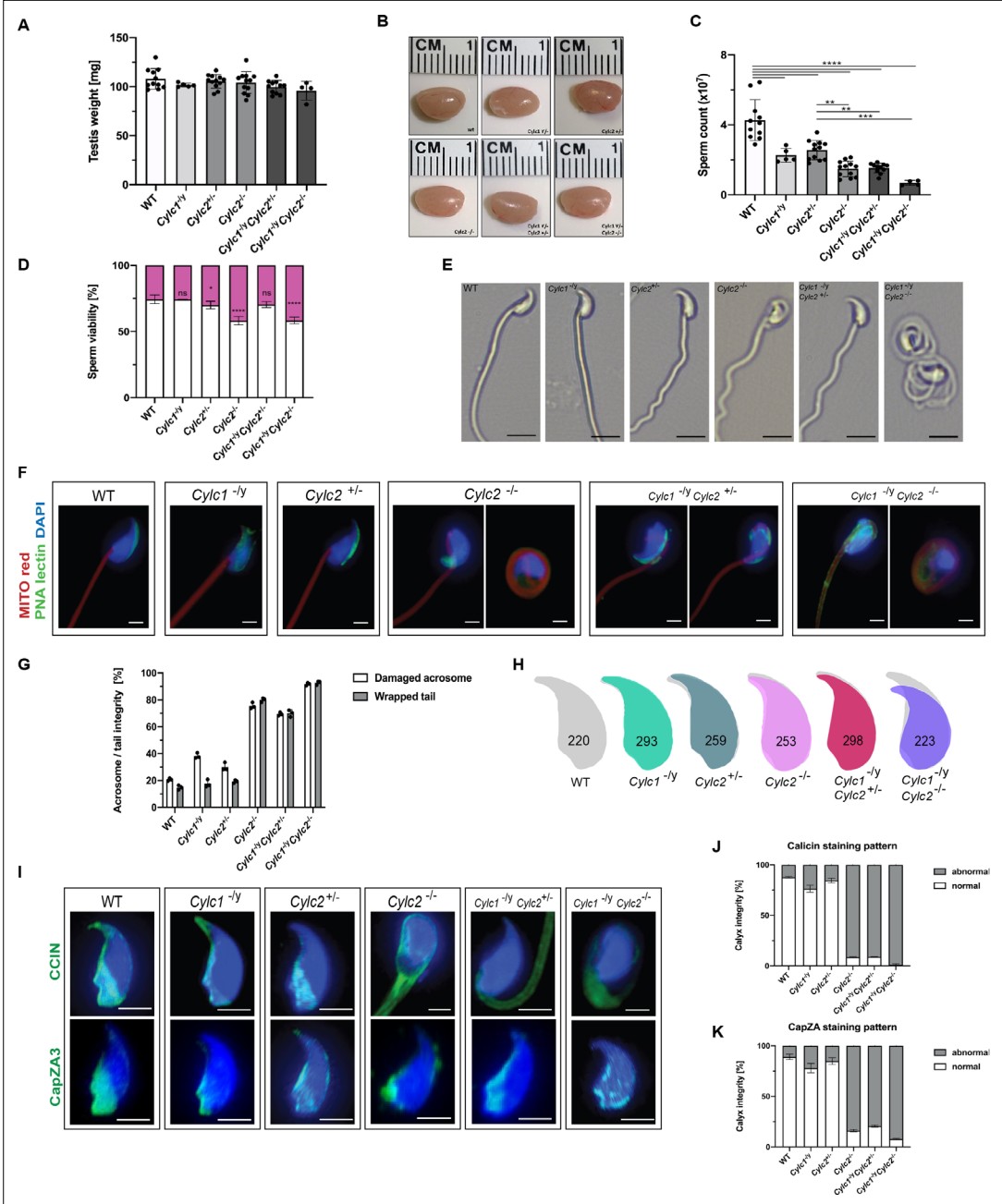

**Figure 2.** Sperm morphology is severely altered in Cylicin-deficient mice. (**A**) Testis weight (mg) and sperm count (×10⁷) of wildtype (WT), *Cylc1⁻/y*, *Cylc2⁺/⁻*, *Cylc2⁻/⁻*, *Cylc1⁻/y Cylc2⁺/⁻*, and *Cylc1⁻/y Cylc2⁻/⁻* males. Mean values ± SD are shown; black dots represent data points for individual males. (**B**) Comparable photographs of the testes of WT, *Cylc1⁻/y*, *Cylc2⁺/⁻*, *Cylc2⁻/⁻*, *Cylc1⁻/y Cylc2⁺/⁻*, and *Cylc1⁻/y Cylc2⁻/⁻* mice. (**C**) Epididymal sperm count (×10⁷) of WT, *Cylc1⁻/y*, *Cylc2⁺/⁻*, *Cylc2⁻/⁻*, *Cylc1⁻/y Cylc2⁺/⁻*, and *Cylc1⁻/y Cylc2⁻/⁻* males. Mean values ± SD are shown; black dots represent data points for individual males. (**D**) Viability of the epididymal sperm stained with Eosin-Nigrosin. Percentage of Eosin negative (viable) and Eosin positive (inviable) sperm is shown. Data represented as mean ± SD. Staining was performed on three animals from each genotype. (**E**) Bright-field microscopy pictures of epididymal sperm from WT, Cylc1⁻/y, Cylc2⁺/⁻, and Cylc2⁻/⁻ mice. Scale bar: 10 µm. (**F**) Immunofluorescence staining of epididymal sperm acrosomes with peanut agglutinin (PNA) lectin (green) and tails with MITOred (red). Nuclei were counterstained with DAPI. Scale bar: 5 µm. (**G**) Quantification of abnormal sperm of WT, *Cylc1⁻/y*, *Cylc2⁺/⁻*, *Cylc2⁻/⁻*, *Cylc1⁻/y Cylc2⁺/⁻*, and *Cylc1⁻/y Cylc2⁻/⁻* mice is shown. Acrosome aberrations and tail coiling were counted separately. Staining was performed on three animals from each genotype. (**H**) Nuclear morphology analysis of WT, *Cylc1⁻/y*, *Cylc2⁺/⁻*, *Cylc2⁻/⁻*, *Cylc1⁻/y Cylc2⁺/⁻*, and *Cylc1⁻/y Cylc2⁻/⁻* sperm. Number of cells analyzed for each genotype

*Figure 2 continued on next page*

*Figure 2 continued*

is shown. (**I**) Representative pictures of immunofluorescent staining against perinuclear theca (PT) proteins CCIN (upper panel) and CAPZa3 (lower panel) in WT, *Cylc1⁻/ʸ*, *Cylc2⁺/⁻*, *Cylc2⁻/⁻*, *Cylc1⁻/ʸ Cylc2⁺/⁻*, and *Cylc1⁻/ʸ Cylc2⁻/⁻* sperm. Nuclei were counterstained with DAPI. Staining was performed on three animals from each genotype. Scale bar: 5 µm. (**J**–**K**) Quantification of sperm with abnormal calyx integrity in WT, *Cylc1⁻/ʸ*, *Cylc2⁺/⁻*, *Cylc2⁻/⁻*, *Cylc1⁻/ʸ Cylc2⁺/⁻*, and *Cylc1⁻/ʸ Cylc2⁻/⁻* mice based on CCIN and CapZA staining patterns.

The online version of this article includes the following source data and figure supplement(s) for figure 2:

**Source data 1.** Testis weights and sperm counts.

**Source data 2.** Sperm viability.

**Source data 3.** Brightfield micrographs of mature sperm.

**Source data 4.** Staining and quantification of acrosomal and flagellar defects.

**Source data 5.** Sperm head morphology.

**Source data 6.** Staining of calyx proteins.

**Figure supplement 1.** Hematoxylin and eosin (HE)-stained testicular tissue sections of wildtype (WT), *Cylc1⁻/ʸ*, *Cylc2⁺/⁻*, *Cylc2⁻/⁻*, *Cylc1⁻/ʸ Cylc2⁺/⁻*, and *Cylc1⁻/ʸ Cylc2⁻/⁻* mice.

**Figure supplement 2.** Eosin-Nigrosin staining of epididymal sperm samples from wildtype (WT), *Cylc1⁻/ʸ*, *Cylc2⁺/⁻*, *Cylc2⁻/⁻*, *Cylc1⁻/ʸ Cylc2⁺/⁻*, and *Cylc1⁻/ʸ Cylc2⁻/⁻* mice.

**Figure supplement 3.** Nuclei of wildtype (WT), *Cylc1⁻/ʸ*, *Cylc2⁺/⁻*, *Cylc2⁻/⁻*, *Cylc1⁻/ʸ Cylc2⁺/⁻*, and *Cylc1⁻/ʸ Cylc2⁻/⁻* sperm stained with DAPI.

**Figure supplement 4.** Co-staining against CYLC1/CYLC2 (red) and CCIN (green) in epididymal sperm cells of wildtype (WT) mouse.

functional domains with repetitive, lysine-lysine-aspartic acid (KKD) and lysine-lysine-glutamic acid (KKE) peptide motifs were depleted. Deletion was confirmed by PCR-based genotyping (*Figure 1B*).

Fertility testing of *Cylc1⁻/ʸ* males revealed significantly reduced pregnancy rates (16%) and mean litter size (2.2) (*Figure 1C*). *Cylc2⁻/⁻* males were infertile, while *Cylc2⁺/⁻* males showed no significant difference in fertility parameters compared to wildtype (WT) mice (*Figure 1C*). Additionally, established mouse lines were intercrossed to generate *Cylc1⁻/ʸ Cylc2⁺/⁻* and *Cylc1⁻/ʸ Cylc2⁻/⁻* males. Of note, *Cylc1⁻/ʸ Cylc2⁺/⁻* and *Cylc1⁻/ʸ Cylc2⁻/⁻* males were infertile (*Figure 1C*). This indicates that loss of *Cylc1* alone is partially tolerated, as suggested by subfertility of *Cylc1⁻/ʸ* males, whereas the additional loss of one *Cylc2* allele renders male mice infertile. Taken together, the results suggest that two functional Cylicin alleles are required for male fertility.

Quantitative reverse transcription-polymerase chain reaction (qRT-PCR) confirmed the absence of *Cylc1* and/or *Cylc2* transcripts in Cylicin-deficient animals (*Figure 1D*). In *Cylc2⁺/⁻* animals expression of *Cylc2* was reduced by 50%. Neither loss of *Cylc1* nor *Cylc2* resulted in upregulation of *Cylc2* or *Cylc1*, respectively.

Next, due to the lack of commercial antisera, polyclonal antibodies against murine CYLC1 and CYLC2 were raised to visualize the localization of Cylicins during spermiogenesis. Specificity of antibodies was proven by immunohistochemical (IHC) stainings, showing a specific signal in testis sections only, but not in any other organ tested (*Figure 1—figure supplement 2*). Immunofluorescence staining of WT testicular tissue showed presence of both, CYLC1 and CYLC2, from the round spermatid stage onward (*Figure 1E*). The signal was first detectable in the subacrosomal region as a cap-like structure, lining the developing acrosome (*Figure 1E–F*, *Figure 1—figure supplement 3*). As the spermatids elongate, CYLC1 and CYLC2 move across the PT toward the caudal part of the cell (*Figure 1—figure supplement 4*). At later steps of spermiogenesis, the localization in the subacrosomal part of the PT faded, while it intensified in the postacrosomal calyx region (*Figure 1E–F*). Of note, the localization of CYLC1 and CYLC2 in the calyx of mature sperm has been reported in bovine and human. The generated antibodies did not stain testicular tissue and mature sperm of *Cylc1*- and *Cylc2*-deficient males, except for a very weak unspecific background staining in the lumen of seminiferous tubules and the residual bodies of testicular sperm (*Figure 1E*). Additionally, western blot analyses confirmed the absence of CYLC1 and CYLC2 in cytoskeletal protein fractions of the respective knockout (*Figure 1G*), thereby demonstrating (i) specificity of the antibodies and (ii) validating the gene knockout. Of note, the CYLC1 antibody detects a double band at 40–45 kDa. This is smaller than the predicted size of 74 kDa, but both bands were absent in *Cylc1⁻/ʸ*. Similarly, the CYLC2 antibody detected a double

band at 38–40 kDa instead of 66 kDa. Again, both bands were absent in $Cylc2^{-/-}$. Next, mass spectrometry analysis of cytoskeletal protein fraction of mature spermatozoa was performed detecting both proteins in WT but not in the respective knockout samples (*Figure 1—figure supplement 5*; *Figure 1—figure supplement 6*).

## Sperm morphology is severely altered in Cylicin-deficient mice

Next, spermiogenesis of Cylicin-deficient males was analyzed in detail. Gross testicular morphology as well as testicular weight were not significantly altered (*Figure 2A and B*). The testicular morphology appeared unaltered, with all stages of spermatogenesis being detectable in hematoxylin and eosin (HE)-stained testicular sections (*Figure 2—figure supplement 1*). However, a strong decline in cauda epididymal sperm counts was observed in all Cylicin-deficient males. For $Cylc1^{-/y}$ and $Cylc2^{+/-}$ males, a moderate reduction of 40–47% was determined, whereas $Cylc2^{-/-}$ and $Cylc1^{-/y} Cylc2^{+/-}$ displayed an approx. 65% reduction in epididymal sperm counts compared to WT mice (*Figure 2C*). In $Cylc1^{-/y} Cylc2^{-/-}$ males, spermiogenesis was most impaired, as indicated by an 85% reduction of the sperm count (*Figure 2C*). Eosin-Nigrosin staining revealed that the viability of epididymal sperm from all genotypes was not severely affected (*Figure 2D*, *Figure 2—figure supplement 2*). However, viability of $Cylc2^{-/-}$ and $Cylc1^{-/y} Cylc2^{-/-}$ sperm was significantly reduced by approx. 15% compared to WT sperm (*Figure 2D*).

Next, we used bright-field microscopy to evaluate the effects of Cylicin deficiency on sperm morphology. Above all, coiling of the sperm tails and kinked sperm heads were observed in $Cylc2^{-/-}$ and $Cylc1^{-/y} Cylc2^{-/-}$ males (*Figure 2E*). To confirm this, we used peanut agglutinin (PNA)-fluorescein isothiocyanite (FITC) lectin immunofluorescence staining to analyze acrosome localization in mature sperm, MITOred, to visualize mitochondria in the flagellum and DAPI to stain the nucleus (*Figure 2F*). Loss of $Cylc1$ alone caused malformations of the acrosome in around 38% of mature sperm, while their flagellum appeared unaltered and properly connected to the head. $Cylc2^{+/-}$ males showed normal sperm tail morphology with around 30% of acrosome malformations. $Cylc2^{-/-}$ mature sperm cells displayed morphological alterations of head and midpiece (*Figure 2F–G*). 76% of $Cylc2^{-/-}$ sperm cells showed acrosome malformations, bending of the neck region, and/or coiling of the flagellum, occasionally resulting in its wrapping around the sperm head in 80% of sperm (*Figure 2F*). While 70% of $Cylc1^{-/y} Cylc2^{+/-}$ sperm showed these morphological alterations, around 92% of $Cylc1^{-/y} Cylc2^{-/-}$ sperm presented with coiled tail and abnormal acrosome (*Figure 2F–G*).

To analyze in detail the sperm head, we used Nuclear Morphology software on DAPI-stained sperm samples. $Cylc1^{-/y}$ and $Cylc2^{+/-}$ sperm showed no alterations of the nuclear shape when compared to WT (*Figure 2H*, *Figure 2—figure supplement 3*). However, heads of $Cylc2^{-/-}$ and $Cylc1^{-/y} Cylc2^{-/-}$ sperm appeared smaller, with shorter hooks and increased circularity of the nuclei as well as reduced elongation (*Figure 2H*, *Figure 2—figure supplement 3*). Interestingly, $Cylc1^{-/y}Cylc2^{+/-}$ sperm heads appeared unaltered, suggesting that only $Cylc2$ has a crucial role for sperm head shaping, and one functional $Cylc2$ allele is sufficient to maintain the correct shape of the nucleus (*Figure 2H*, *Figure 2—figure supplement 3*).

To study the effects of Cylicin deficiency on sperm calyx integrity and morphology, we analyzed the localization of other calyx-specific proteins, such as CCIN and CapZα3. In epididymal sperm, CCIN co-localize with both CYLC1 and CYLC2 in the calyx (*Figure 2—figure supplement 4*). In $Cylc1^{-/y}$ and $Cylc2^{+/-}$ sperm, CCIN localization remained unchanged, being present in the calyx and in the ventral portion of PT as described previously *Zhang et al., 2022b*. However, in 91% of $Cylc2^{-/-}$ sperm, CCIN localized to the tail or in random parts of the sperm head (*Figure 2I and J*). In 91% of $Cylc1^{-/y} Cylc2^{+/-}$ and 98% of $Cylc1^{-/y} Cylc2^{-/-}$ sperm, the localization of CCIN was also significantly altered, with the signal mainly being present in the sperm tail. CapZα3 forms a heterodimer with CapZβ3, creating a functional complex that localizes in the calyx (*Wear and Cooper, 2004*). Immunofluorescence stainings revealed that the localization of CapZα3 remained unchanged in $Cylc1^{-/y}$ and $Cylc2^{+/-}$ mice compared to WT mice. In 84% of $Cylc2^{-/-}$ sperm cells, CapZα3 localized in the caudal portion of the head but without regular calyx localization (*Figure 2I and K*). Interestingly, $Cylc1^{-/y} Cylc2^{+/-}$ mice showed less severe anomalies of the calyx and although CCIN was located almost exclusively in the tail, CapZα3 maintained the correct calyx localization in around 30% of sperm (*Figure 2I*). Finally, 92% of $Cylc1^{-/y} Cylc2^{-/-}$ spermatozoa showed CapZα3 localization across the sperm head without regular calyx shape. These results suggest that the loss of Cylicins impairs the formation of calyx and the

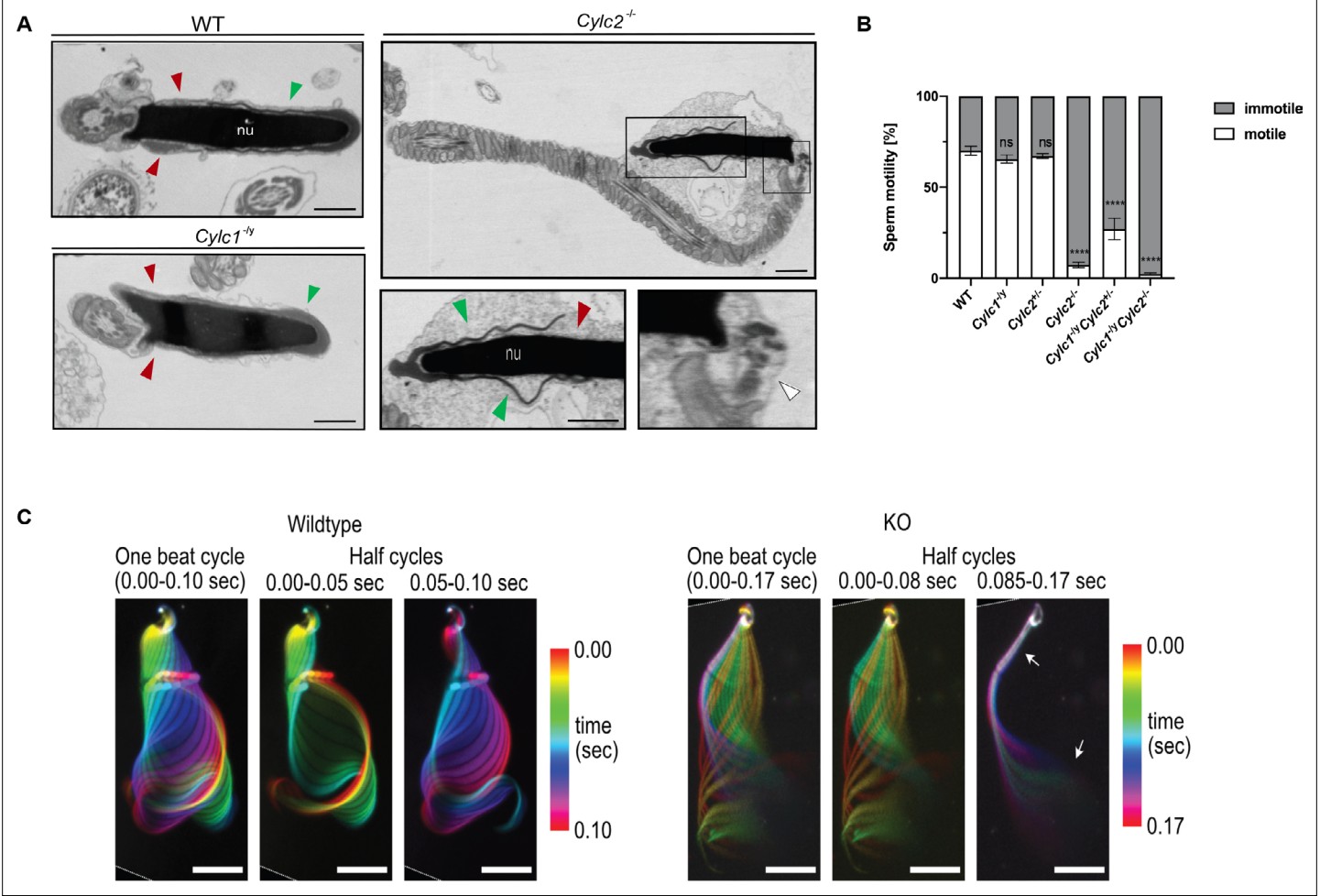

**Figure 3.** *Cylc2⁻ᐟ⁻* sperm cells have altered flagellar beat. (**A**) Transmission electron microscopy (TEM) micrographs of wildtype (WT), *Cylc1⁻ᐟʸ* and *Cylc2⁻ᐟ⁻* epididymal sperm. Acrosome appears detached from the nucleus in *Cylc2⁻ᐟ⁻* sperm (green arrowheads), while the calyx is missing entirely (red arrowheads). The head-tail connecting piece shifted from the basal plate is shown by white arrowheads causing the looping of the flagellum and formation of a cytoplasmatic sac. *Cylc1⁻ᐟʸ* sperm appears comparable to WT. Scale bar: 1 μm. (**B**) Motility of the epididymal sperm of WT, *Cylc1⁻ᐟʸ*, *Cylc2⁺ᐟ⁻*, *Cylc2⁻ᐟ⁻*, *Cylc1⁻ᐟʸ Cylc2⁺ᐟ⁻*, and *Cylc1⁻ᐟʸ Cylc2⁻ᐟ⁻* males activated in TYH medium. (**C**) Full and half-beat cycle plots of the flagellar beat are shown for WT and *Cylc2⁻ᐟ⁻* spermatozoa. Half-beat cycle shows the stiffness of the midpiece (upper arrow) and high oscillations (lower arrow) in *Cylc2⁻ᐟ⁻* sperm in one direction of the beat.

The online version of this article includes the following source data and figure supplement(s) for figure 3:

**Source data 1.** Uncropped TEM-micrographs of mature sperm.

**Source data 2.** Sperm motility.

**Source data 3.** Program codes for flagellar beat analysis.

**Figure supplement 1.** Transmission electron microscopy (TEM) micrographs of wildtype (WT) and *Cylc2⁻ᐟ⁻* sperm and axonemes.

**Figure supplement 2.** SpermQ analysis of the flagellar beat of wildtype (WT) (green) and *Cylc2⁻ᐟ⁻* (red) sperm.

correct localization of its components, which might contribute to morphological anomalies of the sperm described initially.

## *Cylc2⁻ᐟ⁻* sperm cells have altered flagellar beat

Transmission electron microscopy (TEM) of epididymal sperm confirmed the severe structural defects observed by light microscopy (*Figure 3A*, *Figure 3—figure supplement 1*): *Cylc2⁻ᐟ⁻* sperm showed coiling of the tail and dislocation of the head-tail connecting piece from the basal plate, resulting in parallel positioning of head and tail (*Figure 3A*, white arrowheads). Furthermore, in *Cylc2⁻ᐟ⁻* sperm, excess of cytoplasm was observed, located around the nucleus and coiled tail

(*Figure 3A*). Anomalies of the head were observed at the level of the PT, while the nuclei appeared unaltered. In all *Cylc2⁻/⁻* sperm cells, the posterior portion of PT-calyx was absent (*Figure 3A*, red arrowheads). Instead of surrounding the nucleus entirely, the PT in *Cylc2⁻/⁻* sperm appeared interrupted, missing completely its caudal part. Further, we observed loosening of the peri-acrosomal region, which is not compact and adherent to the nucleus (*Figure 3A*, green arrowheads). On the contrary, *Cylc1⁻/y* sperm cells appeared healthy, with intact PT, acrosome, and calyx.

While the motility of *Cylc1⁻/y* and *Cylc2⁺/⁻* sperm remained unchanged compared to WT sperm (around 60% motile cells), motility of *Cylc2⁻/⁻* sperm was drastically reduced to only 7% motile sperm (*Figure 3B*) and the motility of *Cylc1⁻/y Cylc2⁻/⁻* sperm was reduced to 2% motile sperm. In addition, the few motile sperm cells were not progressive but were swimming in circular trajectories. Interestingly, in *Cylc1⁻/y Cylc2⁺/⁻* mice, sperm motility was reduced as well, but less drastically, with 27% of sperm cells being motile (*Figure 3B*).

The SpermQ software was used to analyze the flagellar beat of non-capacitated *Cylc2⁻/⁻* sperm in detail (*Hansen et al., 2018*). *Cylc2⁻/⁻* sperm showed stiffness in the neck and a reduced amplitude of the initial flagellar beat, as well as reduced average curvature of the flagellum during a single beat (*Figure 3—figure supplement 2*). Interestingly, we observed that the flagellar beat of *Cylc2⁻/⁻* sperm cells was similar to WT cells, however, with interruptions during which midpiece and initial principal piece appeared stiff, whereas high-frequency beating occurs at the flagellar tip (*Figure 3C*, *Video 1*, *Video 2*). These interruptions occurred only on the open-hook side and the duration of such interruptions varied from beat to beat. Of note, similar phenotypes have been observed for sperm with structural defects in the axoneme (*Gadadhar et al., 2021*), however axoneme structure of Cylicin-deficient sperm appeared unaltered, presenting typical 9+2 microtubular composition in all genotypes (*Figure 3—figure supplement 1*). Thus, we hypothesize that observed structural defects of the PT and head-tail connecting piece are restrictive for sperm motility and physiological beating patterns.

Taken together, observed anomalies of sperm heads, impaired sperm motility, and the decrease in epididymal sperm concentration show that Cylicin deficiency resembles a severe OAT (oligo-astheno-teratozoospermia syndrome) described in human.

## Cylicins are required for acrosome attachment to the nuclear envelope

To study the origin of observed structural sperm defects, spermiogenesis of Cylicin-deficient males was analyzed in detail. PNA lectin staining and periodic acid Schiff (PAS) staining of testicular tissue sections were performed to investigate acrosome biogenesis. During Golgi phase, the acrosome first starts to appear as an aggregation of proacrosomal vesicles into a single granule. This premature acrosomal structure was unaltered in all genotypes, with PNA signal appearing as a small dot on one pole of round spermatids (*Figure 4A–B*). During cap phase, acrosomes grow to cover the apical part of the nucleus. In WT and *Cylc2⁺/⁻* mice, the forming acrosome appeared equally smooth and showed a regular cap structure on the perinuclear region of round spermatids. However, in some of the round spermatids from *Cylc2⁻/⁻* and *Cylc1⁻/y* mice, gaps in the forming acrosome were observed, as well as an irregular shape of the cap. In *Cylc1⁻/y Cylc2⁺/⁻* and *Cylc1⁻/y Cylc2⁻/⁻* mice, most of the round spermatids were deformed or displayed irregularly localized caps (*Figure 4A–B*, *Figure 4—figure supplement 1*, *Figure 4—figure supplement 2*). At acrosome phase, many elongating spermatids of *Cylc1⁻/y*, *Cylc2⁻/⁻*, *Cylc1⁻/y Cylc2⁺/⁻*, and *Cylc1⁻/y Cylc2⁻/⁻* mice had irregular acrosome (*Figure 4A–B*, *Figure 4—figure supplement 1*, *Figure 4—figure supplement 2*). Detachment of the acrosome from the nuclear envelope was evident in testis samples of *Cylc2⁻/⁻* and *Cylc1⁻/y Cylc2⁻/⁻* male mice. These results suggest that Cylicins are required for the attachment of the developing acrosome to the nuclear envelope during spermiogenesis. Microtubules appeared longer on one side of the nucleus than on the other, displacing the acrosome to the side and creating a gap in the PT (*Figure 4C*). Whereas elongated

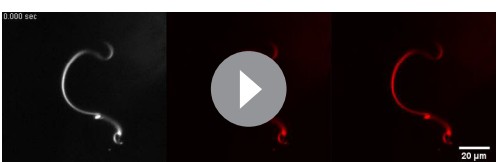

**Video 1.** Full beat cycle of sperm from WT male.
https://elifesciences.org/articles/86100/figures#video1

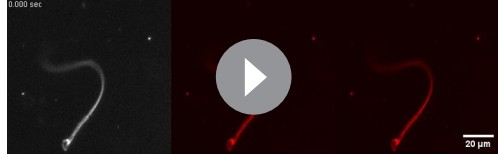

**Video 2.** Full beat cycle of sperm from *Cylc2⁻/⁻* male.
https://elifesciences.org/articles/86100/figures#video2

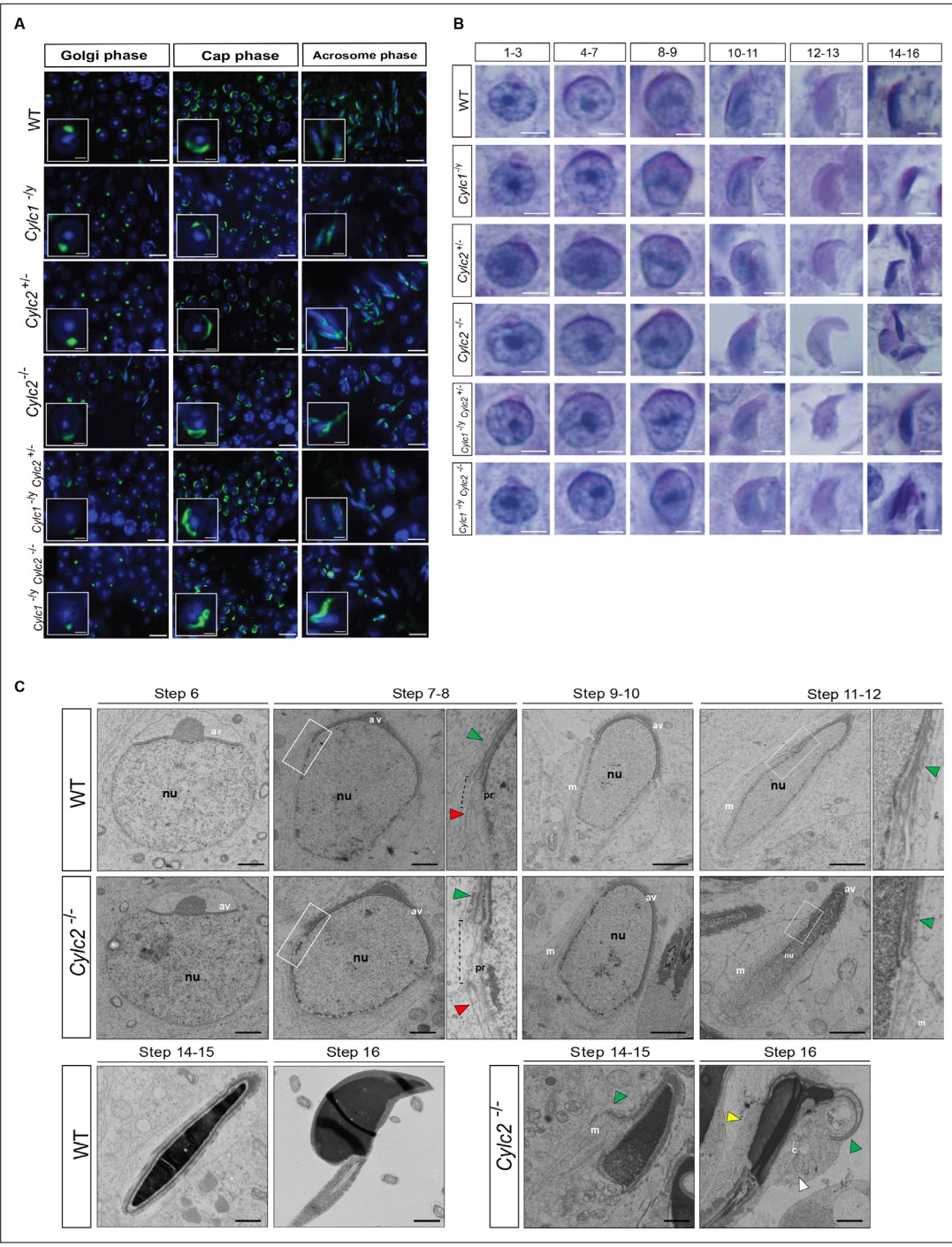

**Figure 4.** Cylicins are required for acrosome attachment to the nuclear envelope. (**A**) Peanut agglutinin (PNA)-fluorescein isothiocyanite (FITC) lectin immunofluorescence staining of the acrosome in testicular tissue of wildtype (WT), $Cylc1^{-/y}$, $Cylc2^{+/-}$, $Cylc2^{-/-}$, $Cylc1^{-/y} Cylc2^{+/-}$, and $Cylc1^{-/y} Cylc2^{-/-}$ mice (green). Golgi phase of acrosome biogenesis at round spermatid stage (I–IV) is visible in the left panel. Middle panel shows cap phases on round spermatids (stage V–VIII). In the right panel acrosomal phase is shown (stage IX–XI). Nuclei were counterstained with DAPI. Staining was performed on three animals from each genotype. Scale bar: 10 μm. Insets show representative single spermatids at higher magnification (scale bar: 2 μm). (**B**) Periodic acid Schiff (PAS) staining of testicular sections from WT, $Cylc1^{-/y}$, $Cylc2^{+/-}$, $Cylc2^{-/-}$, $Cylc1^{-/y} Cylc2^{+/-}$, and $Cylc1^{-/y} Cylc2^{-/-}$ mice. Representative spermatids at different steps of spermiogenesis are shown. Scale bar: 3 μm. (**C**) Transmission electron microscopy (TEM) micrographs of testicular tissues of WT and $Cylc2^{-/-}$ mice. Single spermatids from step 6 to step 16 are shown. nu: nucleus; av: acrosomal vesicle; pr: perinuclear ring; m: manchette microtubules; cy: cytoplasm; green

*Figure 4 continued on next page*

*Figure 4 continued*

arrowheads: developing acrosome; red arrowheads: manchette; white arrowhead: cytoplasm; yellow arrowhead: remaining microtubules in mature sperm. Scale bar: 1 µm.

The online version of this article includes the following source data and figure supplement(s) for figure 4:

**Source data 1.** PNA-lectin stained testes tissues.

**Source data 2.** Uncropped TEM-micrographs of testes tissues.

**Figure supplement 1.** Peanut agglutinin (PNA)-lectin immunofluorescence staining of wildtype (WT), *Cylc1*<sup>-/y</sup>, *Cylc2*<sup>+/-</sup>, *Cylc2*<sup>-/-</sup>, *Cylc1*<sup>-/y</sup> *Cylc2*<sup>+/-</sup>, and *Cylc1*<sup>-/y</sup> *Cylc2*<sup>-/-</sup> testicular tissue.

**Figure supplement 2.** Periodic acid Schiff (PAS)-stained testicular tissue sections of wildtype (WT), *Cylc1*<sup>-/y</sup>, *Cylc2*<sup>+/-</sup>, *Cylc2*<sup>-/-</sup>, *Cylc1*<sup>-/y</sup> *Cylc2*<sup>+/-</sup>, and *Cylc1*<sup>-/y</sup> *Cylc2*<sup>-/-</sup> mice.

**Figure supplement 3.** Transmission electron microscopy (TEM) micrographs of degrading damaged spermatids in testicular sections of *Cylc2*<sup>-/-</sup> mice.

spermatids at steps 14–15 in WT sperm already disassembled their manchette and the PT appeared as a unique structure that compactly surrounds the nucleus, in *Cylc2*<sup>-/-</sup> spermatids, remaining microtubules failed to disassemble (*Figure 4C*, yellow arrowhead), and the acrosome detached from the nuclear envelope (green arrowhead). In addition, at step 16, the calyx was absent, and an excess of cytoplasm surrounded the nucleus and flagellum (*Figure 4C*, white arrowhead). Furthermore, many damaged and degrading cells were observed in testicular tissue TEM samples, having perforated nuclei and detached structures (*Figure 4—figure supplement 3*). Interestingly, phagosomes with cellular remains were observed far away from the lumen and sometimes even at the basal membrane of the tubuli, suggesting that the cells that suffer most severe structural damage are being removed. This mechanism of removing malformed cells explains the reduction of epididymal sperm count in Cylicin-deficient genotypes.

## Cylicin deficiency results in abnormal manchette elongation and disassembly

Next, we investigated the role of Cylicins during formation and development of the manchette – a sperm-specific, transient structure that represents a microtubular platform for protein transport, which showed several anomalies in TEM. Transport of the intracellular vesicles is crucial for the formation of the flagellum, acrosome assembly, and removal of excess cytoplasm. The manchette is first detected at step 8 at the perinuclear ring of round spermatids, just prior to their elongation. During the next steps of spermiogenesis, as the spermatids elongate, manchette moves toward the neck region in a skirt-like structure and starts disassembling at step 13 when the elongation is complete (*Okuda et al., 2017*). We used immunofluorescence staining of α-tubulin on squash testis samples containing spermatids at different stages of spermiogenesis to investigate whether the altered head shape, calyx structure, and tail-head connection anomalies originate from defects of the manchette structure. Spermatids starting from step 8 were observed individually for step-to-step comparison. In all genotypes, a cap-like shape of the manchette was observed in step 8 round spermatids, suggesting that the manchette assembles properly and starts elongating toward the neck region during step 9 (*Figure 5—figure supplement 1*). In all genotypes, the typical skirt-like structure was observed at the caudal region of the spermatids at steps 10 and 11, suggesting that the manchette assembles correctly even in Cylicin-deficient sperm (*Figure 5A*). In spermatids from *Cylc1*<sup>-/y</sup> and *Cylc2*<sup>+/-</sup> mice, regular manchette development was observed in further steps of spermiogenesis (*Figure 5A*). However, starting from step 12, spermatids from *Cylc2*<sup>-/-</sup>, *Cylc1*<sup>-/y</sup>*Cylc2*<sup>+/-</sup>, and *Cylc1*<sup>-/y</sup>*Cylc2*<sup>-/-</sup> mice showed abnormal manchette elongation, which became more prominent at step 13 (*Figure 5A*). Manchette length was measured starting from step 10 until step 13 spermatids and the mean was obtained, showing that the average manchette length was 76–80 nm in WT, *Cylc1*<sup>-/y</sup> and *Cylc2*<sup>+/-</sup>, while for *Cylc2*<sup>-/-</sup> and *Cylc1*<sup>-/y</sup>*Cylc2*<sup>-/-</sup> spermatids mean manchette length reached 100 nm (*Figure 5B*). *Cylc1*<sup>-/y</sup>*Cylc2*<sup>+/-</sup> spermatids displayed an intermediate phenotype with a mean manchette length of 86 nm. Interestingly, some of *Cylc2*-deficient spermatids showed shifting of the manchette to the ventral side of the nucleus along with excessive elongation. At step 16, the manchette was normally disassembled in WT, *Cylc1*<sup>-/y</sup>, and *Cylc2*<sup>+/-</sup> spermatids (*Figure 5A*). However, *Cylc2*<sup>-/-</sup>, *Cylc1*<sup>-/y</sup>*Cylc2*<sup>+/-</sup>, and *Cylc1*<sup>-/y</sup>*Cylc2*<sup>-/-</sup> spermatids showed a persistent α-tubulin signal, indicating that disassembly of the manchette is delayed or incomplete (*Figure 5A*).

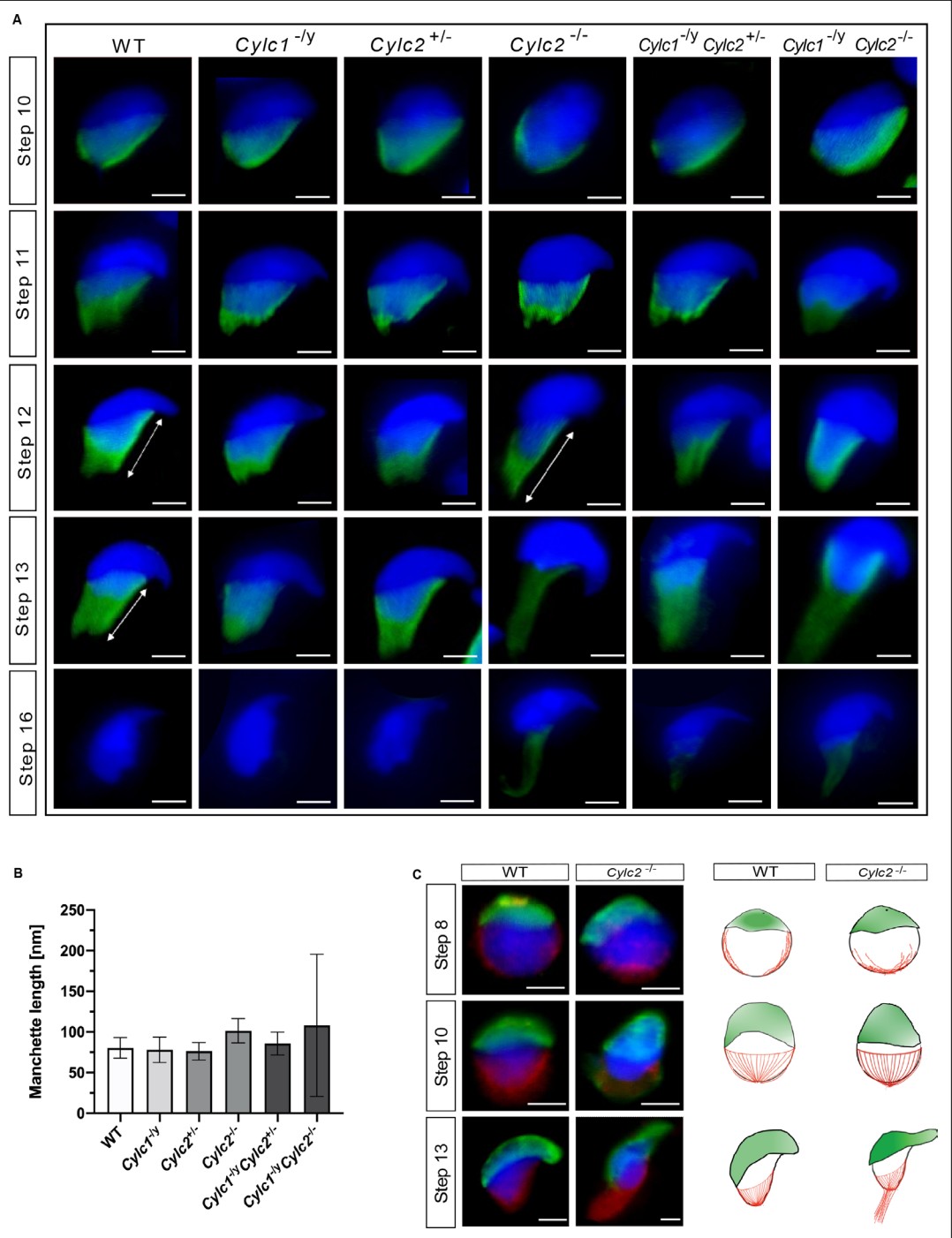

**Figure 5.** Cylc2 deficiency causes delay in manchette removal. (**A**) Immunofluorescence staining of α-tubulin to visualize manchette structure in squash testis samples of wildtype (WT), *Cylc1⁻/ʸ*, *Cylc2⁺/⁻*, *Cylc2⁻/⁻*, *Cylc1⁻/ʸ Cylc2⁺/⁻*, and *Cylc1⁻/ʸ Cylc2⁻/⁻* mice. Spermatids in different steps of spermiogenesis were shown, for step-to-step comparison. Scale bar: 5 µm. (**B**) Quantification of manchette length in WT, *Cylc1⁻/ʸ*, *Cylc2⁺/⁻*, *Cylc2⁻/⁻*, *Cylc1⁻/ʸ Cylc2⁺/⁻*, and *Cylc1⁻/ʸ Cylc2⁻/⁻* α-tubulin-stained spermatids at steps 10–13. (**C**) Co-staining of the manchette with HOOK1 (red) and acrosome with peanut agglutinin (PNA)-lectin (green) is shown in round, elongating and elongated spermatids of WT (upper panel) and *Cylc2⁻/⁻* mice (lower panel). Scale bar: 3 µm. Schematic representation shows acrosomal structure (green) and manchette filaments (red).

The online version of this article includes the following source data and figure supplement(s) for figure 5:

**Source data 1.** Manchette length.

*Figure 5 continued on next page*

*Figure 5 continued*

**Source data 2.** Uncropped PNA-Hook1 staining.

**Figure supplement 1.** Immunofluorescence staining of α-tubulin in wildtype (WT), *Cylc1⁻ᐟʸ*, *Cylc2⁺ᐟ⁻*, *Cylc2⁻ᐟ⁻*, *Cylc1⁻ᐟʸ Cylc2⁺ᐟ⁻*, and *Cylc1⁻ᐟʸ Cylc2⁻ᐟ⁻* squash testis samples.

Other than α-tubulin, we also used HOOK1 as manchette marker. HOOK1 is a member of a family of coiled-coil proteins, which bind to microtubules and organelles and regulate microtubule trafficking during endocytosis and spermiogenesis. Co-staining of the spermatids with antibodies against PNA lectin (green) and HOOK1 (red) revealed that abnormal manchette elongation and acrosome anomalies simultaneously occurred in elongating spermatids of *Cylc2⁻ᐟ⁻* male mice (*Figure 5C*). Schematic representation shows acrosome biogenesis and manchette development in WT and *Cylc2⁻ᐟ⁻* spermatids (*Figure 5C*). While round spermatids of *Cylc2⁻ᐟ⁻* mice elongated as those of the WT sperm, the manchette elongated abnormally and the acrosome became loosened (*Figure 4C*, *Figure 5C*).

## *Cylc2* coding sequence is slightly more conserved among species than *Cylc*1

To address why *Cylc2* deficiency causes more severe phenotypic alterations than *Cylc1* deficiency in mice, we performed evolutionary analysis of both genes. Analysis of the selective constrains on *Cylc1* and *Cyvideolc2* across rodents and primates revealed that both genes' coding sequences are conserved in general, although conservation is weaker in *Cylc1* trending toward a more relaxed constraint (*Figure 6*). A model allowing for separate calculation of the evolutionary rate for primates and rodents did not detect a significant difference between the clades, neither for *Cylc1* nor for *Cylc2*, indicating that the sequences are equally conserved in both clades.

To analyze the selective pressure across the coding sequence in more detail, we calculated the evolutionary rates for each codon site across the whole tree. According to the analysis, 34% of codon sites were conserved, 51% under relaxed selective constraint, and 15% positively selected. For *Cylc2*, 47% of codon sites were conserved, 44% under neutral/relaxed constraint, and 9% positively selected. Interestingly, codon sites encoding lysine residues, which are proposed to be of functional importance for Cylicins, are mostly conserved. For *Cylc1*, 17% of lysine residues are significantly conserved and 35% of significantly conserved codons encode for lysine. For *Cylc2*, this pattern is even more pronounced with 27.9% of lysine codons being significantly conserved and 24.3% of all conserved sites encoding for lysine (*Figure 6*).

## Cylicins are required for normal sperm morphology in human

As loss of two Cylicin alleles causes fertility defects in mice, we next addressed whether infertile men also display variants in *CYLC1/CYLC2*.

Exome sequencing within the MERGE (**M**al**e R**eproductive **G**enomics study) cohort identified one patient (M2270) carrying rare (MAF <0.01, gnomAD) missense variants in both *CYLC1* and *CYLC2*. The man of German origin presented at age of 40 years for couple infertility because unsuccessfully trying to conceive for 6 years. The couple underwent one ICSI procedure which resulted in 17 fertilized oocytes out of 18 retrieved. Three cryo-single embryo transfers were performed in spontaneous cycles, but no pregnancy was achieved.

Patient M2270 carries the hemizygous variant c.1720G>C in *CYLC1* that leads to an amino acid exchange from glutamic acid to glutamine (p.(Glu574Gln)), is predicted to be deleterious or possibly damaging by in silico tools (SIFT [*Ng and Henikoff, 2003*] and PolyPhen [*Adzhubei et al., 2010*], respectively), and has a CADD score of 11.91. It is located in exon 4 out of 5 and affects a region that is predicted to be intolerant to such substitutions (*Figure 7—figure supplement 1*, metadome). It occurs only twice in the gnomAD database (v2.1.1) comprising 141,456 individuals (67,961 XY), once identified in a hemizygous male and once in a female carrier and is absent from our database.

M2270 further carries the heterozygous variant c.551G>A in *CYLC2* that is predicted to be tolerated (SIFT) or benign (PolyPhen) in accordance with a low CADD score of 0.008. It is located in exon 5 out of 8 and affects a region in which variants are likely to be tolerated (*Figure 7—figure supplement 1*, metadome). However, it is a rare variant occurring with an allele frequency of 0.0035 in the general population, according

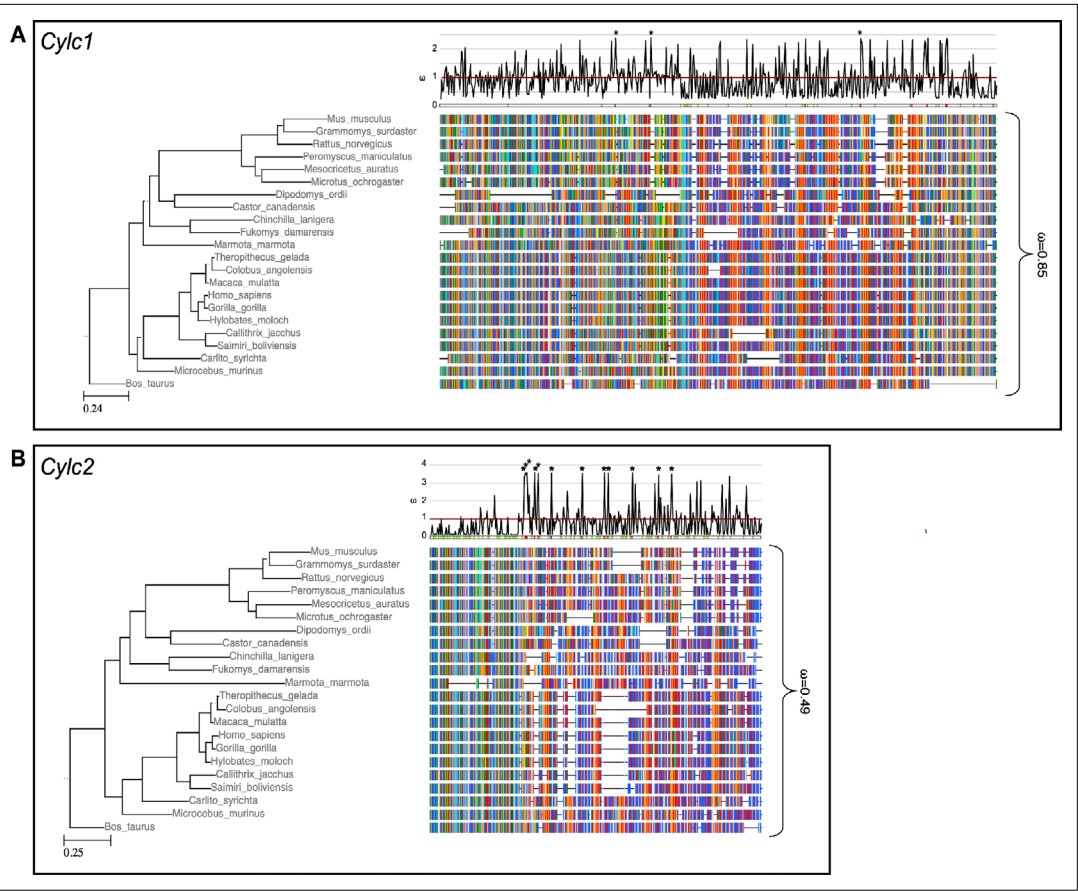

**Figure 6.** Species phylogeny with branch length representing number of nucleotide substitutions per codon with schematic representation of (**A**) CYLC1 and (**B**) CYLC2 amino acid alignment used in the PAML CodeML analysis. Alignments were stripped of columns with gaps in more than 80% of species. Evolutionary rate ($\omega$) obtained by CodeML models M0 is shown for the whole alignment. The graph on top shows the evolutionary rate ($\omega$) per codon sites across the whole tree (CodeML model M2a). Significantly positively selected sites are marked by asterisks. Sites with a probability of higher than 0.95 to belonging to the conserved or positively selected site class are marked in green and red respectively below the graph.

The online version of this article includes the following source data for figure 6:

**Source data 1.** Code ML results table.

to the gnomAD database. Importantly, only three XX individuals are reported to be homozygous for the variant within *CYLC2*.

Segregation analyses revealed maternal inheritance of the X-linked *CYLC1* variant c.1720G>C p. (Glu574Gln), whereas the father carries the heterozygous *CYLC2* variant c.551G>A p.(Gly184Asp) (***Figure 7A***). According to ACMG-AMP criteria (***Richards et al., 2015***) both variants are classified as variants of uncertain significance. No other potentially pathogenic variants in genes associated with sperm morphological defects were identified by exploring the exome data of M2270.

Semen analysis performed following WHO guidelines (***World Health Organization, 2021***) is shown in ***Table 1***. The sperm concentration in the semen was slightly reduced, while significant reduction of motile spermatozoa (12.5%) was observed. Interestingly, only 2% of the sperm cells appeared morphologically normal, while 98% of sperm showed head defects (***Table 1***). Immunofluorescence staining of CYLC1 revealed that while in healthy donor's sperm, CYLC1 localizes in the calyx, in M2270, CYLC1 labeling was absent (***Figure 7B***). Bright-field microscopy demonstrated that M2270's sperm flagella coiled in a similar manner compared to flagella of sperm from Cylicin-deficient mice. Quantification revealed 57% of M2270 sperm displaying abnormal flagella compared to 6% in the healthy donor (***Figure 7D***). Interestingly, DAPI staining revealed that 27% of M2270 flagella carry cytoplasmatic bodies without nuclei and could be defined as headless spermatozoa (***Figure 7C***, white arrowhead;

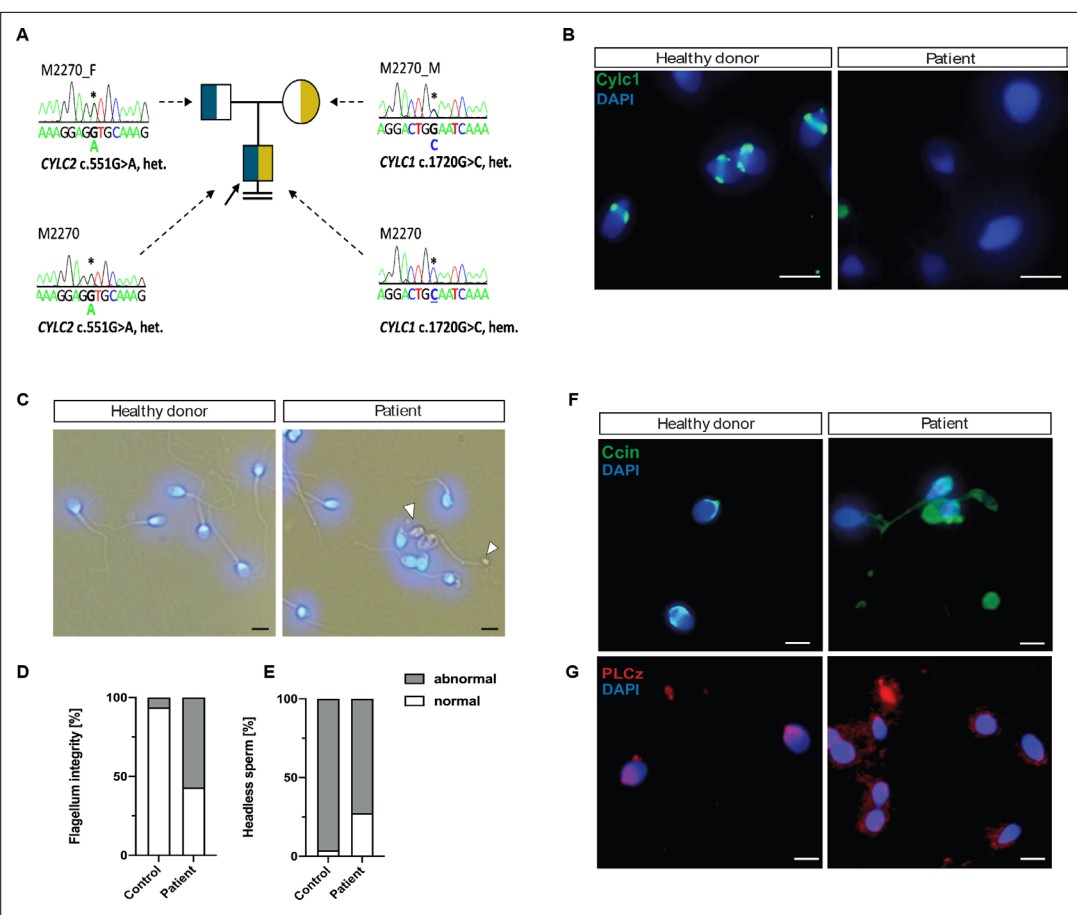

**Figure 7.** Cylicins are required for human male fertility. (**A**) Pedigree of patient M2270. His father (M2270_F) is carrier of the heterozygous *CYLC2* variant c.551G>A and his mother (M2270_M) carries the X-linked *CYLC1* variant c.1720G>C in a heterozygous state. Asterisks (*) indicate the location of the variants in *CYLC1* and *CYLC2* within the electropherograms. (**B**) Immunofluorescence staining of CYLC1 in spermatozoa from healthy donor and patient M2270. In donor's sperm cells CYLC1 localizes in the calyx, while patient's sperm cells are completely missing the signal. Scale bar: 5 µm. (**C**) Bright-field images of the spermatozoa from healthy donor and M2270 show visible head and tail anomalies, coiling of the flagellum, as well as headless spermatozoa, which carry cytoplasmatic residues without nuclei (white arrowhead). Heads were counterstained with DAPI. Scale bar: 5 µm. (**D–E**) Quantification of flagellum integrity (**D**) and headless sperm (**E**) in the semen of patient M2270 and a healthy donor. (**F–G**) Immunofluorescence staining of CCIN (**F**) and PLCz (**G**) in sperm cells of patient M2270 and a healthy donor. Nuclei were counterstained with DAPI. Scale bar: 3 µm.

The online version of this article includes the following source data and figure supplement(s) for figure 7:

**Source data 1.** Uncropped images of stainings on human sperm.

**Figure supplement 1.** Variants in CYLC1 and CYLC2 identified in subject M2270 and their localization on the DNA and protein level.

*Figure 7E*). CCIN staining demonstrated that while spermatozoa of a healthy donor showed a typical, funnel-like calyx structure in the posterior region of PT, spermatozoa from M2270 had CCIN localized in irregular manner throughout head and tail, suggesting that Cylicins have a role in maintenance of the calyx structure and composition in both mice and human spermatozoa (*Figure 7F*). Testis-specific phospholipase C zeta 1 (PLC ζ 1) is localized in the postacrosomal region of PT in mammalian sperm (*Yoon and Fissore, 2007*) and has a role in generating calcium (Ca²⁺) oscillations that are necessary for oocyte activation (*Yoon et al., 2008*). Staining of healthy donor's spermatozoa showed a previously described localization of PLC ζ 1 in the calyx, while sperm from M2270 patient presents signal irregularly through the PT surrounding sperm heads (*Figure 7G*). These results suggest that Cylicin deficiency can cause severe disruption of PT in human sperm as well, causing male infertility.

**Table 1.** Semen analysis of the patient M2770 carrying variants in the *CYLC1* and *CYLC2* genes.

| | First visit | Second visit | WHO reference rang |
|---|---|---|---|
| **Abstinence time** (day) | 4.0 | 5.0 | |
| **Volume** (ml) | 4.2 | 5.8 | >1.4 |
| **Concentration** (Mill./ml) | 10.5 | 16.3 | >16 |
| **Total sperm count** (Mill.) | 44.1 | 94.5 | >39 |
| **Vitality** (%) | 53 | 27 | >54 |
| **Motility** | | | |
| **a** (%) | 7 | 9 | a+b > 30 |
| **b** (%) | 5 | 4 | |
| **c** (%) | 19 | 8 | |
| **d** (%) | 69 | 79 | |
| **Morphology** | | | |
| **Normal** (%) | 2 | 2 | >4 |
| **Head defects** (%) | 99 | 99 | |
| **Midpiece defects** (%) | 63 | 59 | |
| **Flagella defects** (%) | 18 | 47 | |

## Discussion

Spermiogenesis is a highly organized process that is dependent on a unique cytoskeletal organization of the sperm cells. The PT has a role of structural scaffold that surrounds sperm nucleus, and its protein composition is crucial for correct sperm development. In this study, we used CRISPR/Cas9 gene editing to establish *Cylc1*-, *Cylc2*-, and *Cylc1/2*-deficient mouse lines to analyze the role of Cylicins. We demonstrated that the loss of Cylicins impairs male fertility in mice by severely disturbing sperm head architecture. The significance of our findings is supported by the identification of *CYLC1/2* variants in an infertile patient who presents similar structural anomalies in sperm cells.

Detailed analysis of sperm development and morphology demonstrated that infertility of Cylicin-deficient male mice is caused by anomalies of different sperm structures. The prominent loss of PT integrity and calyx structure in the mature sperm lead to acrosome loosening and detachment from the nuclear envelope. Furthermore, the shifting of the basal plate caused damage in the head-tail connecting piece resulting in coiling of the flagellum and impaired swimming. Murine Cylicins are first detected in the peri-acrosomal region of round spermatids and move to the postacrosomal region of PT during spermatid elongation to finally localize to the calyx of mature sperm. This dynamic localization pattern of Cylicins has been described in human, boar, and bovine sperm as well (*Longo et al., 1987*; *Hess et al., 1993*; *Hess et al., 1995*). Interestingly, other PT-enriched proteins CCIN (*Paranko et al., 1988*; *Lécuyer et al., 2000*; *Zhang et al., 2022b*) and CPβ3/CPα3 move in the similar manner across PT during spermiogenesis and are present in the calyx of mature sperm of various mammalian species (*Figure 2—figure supplement 4*). This co-localization suggests the potential interaction between calyx proteins. Furthermore, CCIN (*Longo et al., 1987*; *Paranko et al., 1988*; *von Bülow et al., 1995*) and CPβ3-CPα3 complex (*Bülow et al., 1997*; *Hurst et al., 1998*; *Tanaka et al., 1994*) are described as actin-binding proteins and porcine Cylc2 has been shown to have a high affinity for F-actin as well (*Rousseaux-Prévost et al., 2003*). The potential roles of F-actin during spermiogenesis in mammals involve biogenesis of the acrosome (*Welch and O'Rand, 1985*) and its correct attachment to the outer nuclear membrane of the spermatids (*Russell et al., 1986*) as well as removal of excess cytoplasm (*Russell, 1979*).

The loss of Cylicins caused acrosome detachment from NE starting from cap phase of the acrosome biogenesis. Interestingly, loss of CCIN results in similar loosening of the acroplaxome from the outer nuclear membrane (*Zhang et al., 2022b*). CCIN is shown to be necessary for the IAM-PT-NE complex by establishing bidirectional connections with other PT proteins. Zhang et al. found CYLC1

to be among proteins enriched in PT fraction (*Zhang et al., 2022b*). Based on their speculation that CCIN is the main organizer of the PT, we hypothesize that both CCIN and Cylicins might interact, either directly or in a complex with other proteins, in order to provide the 'molecular glue' necessary for the acrosome anchoring. Furthermore, Cylicin deficiency resulted in cytoplasmatic retention and bending of the midpiece, similarly to the CPα3 mutant phenotype repro32/repro32 (*Geyer et al., 2009*). The CAPZ complex has been shown to regulate the actin dynamics by preventing addition or loss of G-actin subunits to the ends of F-actin filaments (*Wear et al., 2003*). Since we demonstrated that the loss of Cylicins resulted in mislocalization of both CCIN and CPα3, we speculate that Cylicins have a crucial role in maintenance of the integrity of PT structure and thus are required for proper function of PT proteins.

Other than morphological defects of the mature sperm PT, during spermiogenesis Cylicin deficiency results in excessive manchette elongation, its delayed disassembly, as well as formation of abnormal gaps in the PT at the level of perinuclear ring. The intra-manchette transport (IMT) of proteins from the apical pole of the head to the base of the developing tail occurs during spermiogenesis (*Lehti and Sironen, 2016*). The malfunctions of the IMT can cause delay in manchette clearance and morphological defects of mature sperm. Mouse models deficient for IMT proteins such as HOOK1 (*Mendoza-Lujambio et al., 2002*), CEP131 (*Hall et al., 2013*), and IFT88 (*Kierszenbaum et al., 2011*) show abnormal manchette elongation and delay in its clearance resulting in aberrant nuclear shape of the sperm. Furthermore, the loss of HOOK1 results in head-tail connection and midpiece anomalies such as flipping of the head, basal plate defects, and coiling of the tail similar to *Cylc2* deficiency (*Mochida et al., 1999*). The localization of manchette on the caudal portion of spermatids coincides with the localization of the calyx in the mature sperm, so these results indicate that the manchette might be maintained longer to compensate for the missing formation of the calyx structure. Furthermore, we observed wide gaps in the perinuclear ring during manchette elongation, suggesting that Cylicins might have a role in maintenance of the contact between caudal and apical PT region and its integrity.

Our evolutionary analysis of *Cylc1* and *Cylc2* genes across rodents and primates revealed that both coding sequences are under purifying selection. Overall, the results reveal that *Cylc1* is under slightly less conserved constraint than *Cylc2* leading to the hypothesis that loss of function in *Cylc1* might be less severe and could be compensated for by *Cylc2* due to partial redundancy. This hypothesis is supported by our finding that *Cylc1* deficiency causes subfertility in male mice, while the loss of both *Cylc2* alleles results in male infertility. Furthermore, *in Cylc2*+/- male mice fertility was preserved, while *Cylc1*-/y*Cylc2*+/- males were unable to sire offspring leading to the conclusion that the loss of one *Cylc2* allele could be compensated by *Cylc1*, however at least two functional Cylicin alleles are required for male fertility in mice. Interestingly, *Cylc1*-/y*Cylc2*+/- males displayed an 'intermediate' phenotype, showing slightly less damaged sperm than *Cylc2*-/- and *Cylc1*-/y*Cylc2*-/- animals. This further supports our notion that loss of the less conserved *Cylc1* gene might be at least partially compensated by the remaining *Cylc2* allele.

In general, the evolutionary rate of C-terminal lysine-rich region of both Cylicins seems to be highly volatile between conserved and positively selected codon sites, while the lysine residues seem to be strongly conserved. Changes in the C-terminal region, potentially affecting the length of the lysine-rich domain, might have an adaptive advantage. Targeted positive selection on codon sites could also be a signature of co-evolution with a rapidly evolving interactor.

Sperm morphological defects and infertility observed in one patient with variants in both Cylicin genes point toward a requirement for human spermiogenesis and fertility. A defect in acrosome formation and the existence of variants in two out of three alleles in both *CYLC* genes is in line with the observations made in mice. The absence of CYLC1 confirmed through immunofluorescent staining indicates a functional impact of the missense variant on the CYLC1 protein. Furthermore, impaired CCIN localization was observed in patient M2270 as well, suggesting that PT has similar roles in human and rodents despite the differences in sperm head shape (*Courtot, 1991*). These results suggest that establishing mouse models deficient for Cylicin and other PT proteins might provide insights into mechanisms of human spermiogenesis and cytoskeletal regulation as well. However, with our data we cannot exclude the possibility that there is a discrepancy between mice and men and that the missense variant in *CYLC1* might alone be sufficient to cause the observed phenotype of M2270. The o/e ratios of both genes calculated within the gnomAD database rather indicate a stronger selective pressure on human *CYLC1* (0.08) than on *CYLC2* (0.84). Furthermore, the *CYLC1* variant affects an

**Table 2.** Protospacer sequences.

| Name | Protospacer sequence (5'–3') |
| --- | --- |
| *Mm.Cas9.CYLC1.sg1* | GGTTTATCCATACGTGAGT |
| *Mm.Cas9.CYLC1.sg2* | GGCTTAGGTGATGCTCTAAA |
| *Mm.Cas9.CYLC2.1.AB* | AAGGGAGAGTCGAAAAGCGT |
| *Mm.Cas9.CYLC2.1.AF* | GGATCCAAGGATAAAGTGTC |

intolerant region within the protein sequence according to metadome (*Wiel et al., 2019*), whereas the *CYLC2* variant is located in a region that is more tolerant to variation (*Figure 7—figure supplement 1*). Therefore, we cannot definitively validate the hypothesis of an oligogenic disease in men as well.

The *CYLC2* missense variant is inherited by the father of M2270 (*Figure 5A*) and, thus, not sufficient to cause infertility. However, the father reported difficulties to conceive naturally. Based on this one family, we cannot exclude an effect of pathogenic heterozygous *CYLC2* variants.

Overall, the identification and detailed characterization of further patients with variants in *CYLC1* and *CYLC2* is warranted to draw firm conclusions on the effect of variants in these genes on spermiogenesis and infertility.

## Methods

### Animals

All animal experiments were conducted according to German law of animal protection and in agreement with the approval of the local institutional animal care committees (Landesamt für Natur, Umwelt und Verbraucherschutz, North Rhine-Westphalia, approval IDs: AZ84-02.04.2013.A429, AZ81-02.04.2018.A369). *Cylc1-* and *Cylc2*-deficient mice were generated by CRISPR/Cas9-mediated gene editing in zygotes of the hybrid strain B6D2F1. Guide sequences are depicted in *Table 2*. For *Cylc1*, in vitro transcribed sgRNAs (50 ng/µl each) and Cas9 mRNA (100 ng/µl) (Sigma-Aldrich, Taufkirchen, Germany) were microinjected into the cytoplasm of zygotes as described previously (*Schneider et al., 2016*). *Cylc2*-deficient mice were generated by electroporation of ribonucleoprotein (RNP) complexes into zygotes using a GenePulser II electroporation device (Bio-Rad, Feldkirchen, Germany). For RNP formation, crRNA and tracrRNA (IDT, Leuven, Belgium) were combined in duplex buffer (IDT) to a final concentration of 50 mM each and annealed (95°C, 5 min; cool down to room temperature with –0.2 °C/s). For RNP assembly, 4 µM Cas9 protein (IDT) and 2 µM of each annealed cr/tracr RNA were combined in Opti-MEM media (Gibco 11058-021, Thermo Fisher, Carlsbad, CA, USA) and incubated for 10 min at room temperature. RNP complexes were diluted 1:2 in Opti-MEM, supplemented with 30–40 zygotes in a 0.1 cm gene-pulser cuvette (Bio-Rad) and electroporated (two 30 V square wave pulses, 2 ms pulse length, 100 ms pulse interval). Recovered embryos were cultured over night at 37°C, 5% $CO_2$ in G-TL medium (Vitrolife, Göteborg, Sweden) and transferred into the oviduct of pseudopregnant CB6F1 foster mice at two-cell stage. Offspring were genotyped and gene-edited alleles were separated by backcrossing of founder animals with C57Bl/6J mice. Established mouse lines were registered with Mouse Genome Informatics: Cylc1[em1Hsc] (MGI: 7341368), Cylc2[em1Hsc] (MGI:6718280). Mouse lines were maintained as congenic strains on C57Bl/6J background. For all analyses sexually mature males at the age of 8–15 weeks, backcross generation ≥N3 were used.

### Genomic DNA extraction and genotyping

Genomic DNA was extracted from biopsies using the HotShot method (*Truett et al., 2000*). PCRs were assembled according to the manufacturer's protocol of the DreamTaq Green DNA Polymerase (Thermo Fisher, EP0712) using gene-specific primers listed in *Table 3*.

### Fertility analysis

Males were mated 1:2/1:1 with C57Bl/6J females, which were checked daily for presence of a vaginal plug indicative for successful copulation. Plug positive females were separated and pregnancies as well as litter size were recorded.

## Sampling

Mature sperm were obtained from the cauda epididymis in M2 medium (Merck) or PBS preheated at 37°C. Following several incisions of the cauda, sperm were retrieved by flush-out for 15–30 min. The extracted sperm samples were washed in PBS, centrifuged (4000 rpm, 5 min, 4°C), and snap-frozen. For squash tubule samples, fresh testes from three animals of each genotype were transferred to PBS and tubules were dissected as described by *Kotaja et al., 2006*. Tubule sections were pressed on a slide, quickly frozen in liquid nitrogen, and fixed in 90% ethanol for 5 min.

## Semen analysis

Sperm concentrations for at least four animals of each genotype were determined using a Neubauer hemocytometer. Viability of sperm was determined by Eosin-Nigrosin staining as described previously (*Schneider et al., 2020*) for at least three animals per genotype. Sperm motility was analyzed referring to the WHO guidelines for analysis of human semen. For all analyses at least 100 sperm per individual were analyzed and the percentage of motile/immotile, viable/inviable sperm was calculated.

## Quantitative reverse transcription-polymerase chain reaction

RNA was extracted from the testis tissue with TRIzol (Life Technologies, Carlsbad, CA, USA; 15596018). RNA concentrations and purity ratios were determined by NanoDrop ONE (Thermo Scientific) measurements. qRT-PCR was performed as described previously (*Courtot, 1991*), on ViiA 7 Real Time PCR System (Applied Biosystems) using Maxima SYBR Green qPCR Mastermix (Thermo Fisher; K0221). Replicate of 3 was used for each genotype. Beta-actin was used as a housekeeping gene for normalization. Primers used are shown in *Table 4*.

## Subcellular protein extraction and western blot analysis

For the extraction of cytoskeletal protein fraction from sperm cells, Subcellular Fractionation Kit for Cultured cells (Thermo Fisher, #78840) was used according to the manufacturer's instructions, with slight modifications to adjust it to sperm cells. Five subcellular protein fractions were extracted: membrane protein fraction, cytoplasmic protein fraction, soluble nuclear fraction, chromatin bound fraction, and cytoskeletal protein fraction. Briefly, epididymal sperm cells were washed in ice-cold PBS and centrifuged at 500 × $g$ for 5 min at 4°C. Pellet was resuspended in Cytoplasmatic Extraction Buffer (CEB), incubated for 10 min and centrifuged again at 500 × $g$ for 5 min. Following steps of subcellular fractionation were performed according to the manufacturer's instructions. Quantity of proteins in each fraction was determined using NanoDrop ONE (Thermo Fisher). For further analysis only cytoskeletal protein fraction was used.

Cytoskeletal proteins were separated on 12% SDS gel with 5% stacking gel. Transfer to PVDF membrane was performed using Trans Blot Turbo System (Bio-Rad). Membranes were washed with TBST 1×, stained with Coomassie blue, and blocked with 1% milk for 1 hr at room temperature with gentle shaking. Primary antibodies anti-CYLC1 and anti-CYLC2 were diluted in milk and incubated overnight at 4°C (for antibody dilutions see *Table 5*). After washing in TBST, membranes were incubated for 1 hr at room temperature with polyclonal goat anti-rabbit secondary antibody IgG/HRP (P044801-2; Agilent Technologies/Dako, Santa Clara, CA, USA), diluted 1:2000 in milk. After extensive TBST washing, membranes were imaged using WESTARNOVA2.0 chemiluminescent

**Table 3.** PCR primer sequences.

| Cylc1 | 5'–3' | Expected band size |
| --- | --- | --- |
| *Cylc1_F1* | TATACACACAATCCACAATCTTGAAAT | WT: 427 bp |
| *Cylc1_R1* | TCACTTCAAAAATCCAACTTGTCCT | *KO: 264* bp |
| *Cylc1_R2* | TGCCTAGTATTTCAGGTTCCCC | |
| Cylc2 | | |
| *Cylc2_F1* | ACCACCATTATGGATGCACCG | WT: 376 bp |
| *Cylc2_R1* | AGTGTTTCTTGTGAGCTCGTTG | KO: 286 bp |
| *Cylc2_R2* | GGCTGAATCTTTACCCTTAGGT | |

**Table 4.** qRT primer sequences.

| Name | Forward (5'-3') | Reverse (5'-3') |
|---|---|---|
| *Cylc1* | GGGGAAAAATAAGCTCATGTGTAG | TTCAGGTTCCCCATTGGTTA |
| *Cylc2* | GCATTTCCCAAACCACCAAGG | AACGGATGGTCTCTCGGATATT |
| *Beta-actin* | TGTTACCAACTGGGACGACA | GGGTGTTGAAGGTCTCAAA |

substrate (Cyanagen) or SuperSignal West Femto Maximum Sensitivity Substrate (34095; Thermo Fisher) and ChemiDoc MP Imaging system (Bio-Rad). Membranes were further re-blocked in 3% BSA for 1 hr at room temperature with gentle shaking and incubated with α-tubulin at 4°C overnight. After washing in TBST, membranes were incubated for 1 hr at room temperature with polyclonal rabbit anti-mouse secondary antibody IgG/HRP (P0260; Agilent Technologies/Dako) diluted 1:2000 in 3% BSA.

## Proteomics
### Peptide preparation
All chemicals from Sigma unless otherwise noted.

Cytoskeletal protein solutions extracted as described in previous paragraph were processed with the SP3-approach (*Hughes et al., 2019*). Briefly, protein lysate with 50 µg protein were subjected to cysteine reduction and alkylation with 20 mM DTT and 40 mM acrylamide in 50 mM triethylammonium bicarbonate. Then a mixture of hydrophilic carboxylate-coated magnetic beads (equal amounts of Sera-Mag SpeedBeads, GE Healthcare, cat. no. 45152105050250, and cat. no. 65152105050250) were added at a bead:protein ratio of 10:1 (wt/wt). Protein binding was induced by adding three volumes of ethanol and subsequent mixing for 5 min. Beads with bound proteins were then washed three times with 80% ethanol and finally subjected to overnight tryptic digestion at 37°C using a trypsin:protein ratio of 1:25. Peptide solutions were separated from the magnetic beads, dried in a vacuum concentrator, and stored at –20°C. Before measurements, 10 µg of peptides were further desalted with C18 ZipTips (Merck Millipore, Darmstadt, Germany) to ensure complete removal of beads.

**Table 5.** Antibodies.

| Antibody | Company | Catalogue number | Antigen | Dilution IF | Dilution WB |
|---|---|---|---|---|---|
| α-Tubulin | Abcam | ab7291 | | | 1:10,000 in 3% BSA |
| α-Tubulin | Merck Millipore (Billerica, MA, USA) | 16-232 | | 1:1000 | |
| CapZa3 | Progen | GP-SH4 | | 1:500 | |
| Ccin | Progen | GP-SH3 | | 1:500 | |
| Cylc1 (used for mouse) | Davids Biotechnology (Regensburg, Germany) | Custom-made polyclonal antibody | AESRKSKNDERRKTLKIKFRGK and KDAKKEGKKKGKRESRKKR | 1:1000 (sperm cells) 1:500 (testis tissue) 1:1000 western blot | 1:1000 in 5% milk in TBST |
| Cylc1 (used for human samples) | Santa Cruz | sc-166400 | | 1:500 | |
| Cylc2 | Davids Biotechnology (Regensburg, Germany) | Custom-made polyclonal antibody | KSVGTHKSLASEKTKKEVK and ESGGEKAGSKKEAKDDKKDA | 1:1000 (sperm cells) 1:500 (testis tissue) 1:1000 western blot | 1:1000 in 5% milk in TBST |
| Hook1 | Proteintech | 10871-1-AP | | 1:500 | |
| PLC ζ | Invitrogen | PA5-50476 | | 1:100 | |
| Sp56 | Invitrogen | MA1-10866 | | 1:500 | |

## LC-MS analysis

Dried peptides were dissolved in 10 µl 0.1% formic acid (solvent A). Peptide separation was performed on a Dionex Ultimate nano HPLC system (Dionex GmbH, Idstein, Germany) coupled to an Orbitrap Fusion Lumos mass spectrometer (Thermo Fisher Scientific, Bremen, Germany). One µg peptides were injected onto a C18 analytical column (400 mm length, 100 µm inner diameter, ReproSil-Pur 120 C18-AQ, 3 µm).

The samples were analyzed by a standard data-dependent (DDA) method: Peptides were separated during a linear gradient from 5% to 35% solvent B (90% acetonitrile, 0.1% FA) at 300 nl/min within 120 min. Data-dependent acquisition was performed on ions between 330 and 1600 m/z scanned in the Orbitrap detector every 2.5 s (R=120,000, standard gain control and inject time settings). Polysiloxane (m/z 445.12002) was used for internal calibration. z>1 ions were subjected to higher-energy collision-induced dissociation (1.0 Da quadrupole isolation, threshold intensity 25,000, collision energy 28%) and fragments analyzed in the Orbitrap (R=15,000). Fragmented precursor ions were excluded from repeated analysis for 25 s.

## Data analysis

Raw data processing of DDA data and analysis of database searches were performed with Proteome Discoverer software 2.5.0.400 (Thermo Fisher Scientific). Peptide identification was done with an in-house Mascot server version 2.8.1 (Matrix Science Ltd, London, UK) against the Uniprot reference proteome for *M. musculus* (as of 06/28/23) and a collection of common contaminants (*Frankenfield et al., 2022*). Precursor ion m/z tolerance was 10 ppm, fragment ion tolerance 20 ppm. Tryptic peptides (trypsin/P) with up to two missed cleavages were searched, propionamide was set as a static modification of cysteines, while oxidation of methionine and acetylation of protein N-termini were set as dynamic modifications. Spectrum confidence of Mascot results was assessed by the Percolator algorithm 3.05 as implemented in Proteome Discoverer (*Käll et al., 2008*). Spectra without high confident matches (q-value >0.01) were sent to a second-round Mascot search with semi-specific enzyme cleavage and changing the modification of cysteines with propionamide to dynamic. Proteins with two unique proteins in the protein group were reported. For quantification, summed abundances were normalized on total protein amount in Proteome Discoverer.

## High-resolution microscopy of the flagellar beat

Image sequences of mouse sperm were acquired using dark field at an inverted microscope (IX71; Olympus, Hamburg, Germany), equipped with a dark-field condenser and a high-speed camera (ORCA-Flash4.0 V3, C13220-20 CU, Hamamatsu, Hamamatsu City, Japan). A 10× objective (NA 0.4, UPlanFL; Olympus, Hamburg, Germany) with an additional ×1.6 magnifying lens (Olympus, Hamburg, Germany) that was inserted into the light path (final magnification: ×16) was applied. Image sequences were recorded at a rate of 200 frames per second (fps). A custom-made observation chamber was used (*Hansen et al., 2018*). Sperm samples were diluted in THY buffer shortly before insertion of the suspension into the observation chamber. Three WT and three *Cylc2*⁻/⁻ animals were used.

## Sperm nuclear morphology

For the analysis of sperm nuclear morphology, epididymal sperm samples from three animals of each genotype were fixed in methanol and acetic acid (3:1). The samples were spread onto a slide and stained with 4',6-diamidino-2-phenylindole (DAPI) containing mounting medium (ROTImount FluorCare DAPI (Carl Roth GmbH, Karlsruhe, Germany; HP20.1)). The sperm cells were imaged at ×100 magnification, using a Leica DM5500 B fluorescent microscope. At least 200 pictures were taken from each group and analyzed using Nuclear Morphology software (*Skinner et al., 2019*) according to the developer's instructions. The minimum detection area was set to 1.000 pixels while the maximum detection area was 7.000 pixels.

## Histology

Bouin fixed testis tissues were paraffinized, embedded, and sectioned at 3–5 µm using microtome. For histological analysis, the sections were deparaffinized, hydrated incubated with periodic acid (0.5%) for 10 min, rinsed with $H_2O$, and incubated for 20 min with Schiff reagent. Alternatively, tissue sections were stained with Hemalum solution acid (Henricks and Mayer) and Eosin Y solution (Carl Roth).

In both procedures, stained sections were dehydrated in alcohol row and mounted using Entellan (Sigma-Aldrich). Slides were imaged at ×20 and ×63 magnification under bright field using 5500 B microscope.

## Immunofluorescence/immunohistochemistry

Bouin fixed testis tissue sections were deparaffinized in xylene and rehydrated in decreasing alcohol to be used for IHC staining. Squash testis samples fixed in 90% EtOH were used for the staining of α-tubulin and HOOK1. For sperm immunofluorescence, mature sperm cells isolated from cauda epididymis were fixed with methanol acetic acid (3:1), dropped on glass slides, and air dried. After washing in PBS twice, all samples were permeabilized using 0.1% Triton X-100 for 10 min at room temperature. The samples were then blocked with 5% BSA for 30 min, followed blocking with normal horse serum (Vectorlabs, Burlingame, CA, USA; DI-1788) for 30 min at room temperature. For tissue sections, heat-activated antigen retrieval was performed using citrate buffer (pH 6.0). All primary antibodies were incubated overnight at 4°C. Antibodies and dilutions are listed in *Table 5*. The respective secondary antibodies were incubated for 1 hr at room temperature using VectaFluor Labeling Kit DyLight 488 and DyLight 594 (Vectorlabs, Burlingame, CA, USA; DI-1788, DI-1794). Slides were mounted with DAPI containing mounting medium (ROTImount FluorCare DAPI, Carl Roth; HP20.1).

For IHC staining against CYLC1 and CYLC2, after antigen retrieval and blocking procedures, slides were treated with 6% $H_2O_2$ for 30 min. Slides were then incubated with primary antibodies overnight at 4°C. Biotinylated goat anti-rabbit IgG was used as a secondary antibody and incubated for 1 hr at room temperature. Slides were then processed using Vectastain ABC-AP Kit (Vector Laboratories, AK-5001) and stained using ImmPACT Vector Red Substrate Kit, Alkaline Phosphatase (Vector Laboratories, SK-5105) according to the manufacturer's instructions. Counterstaining was performed using hematoxylin.

For the analysis of acrosome biogenesis, PNA-FITC Alexa Fluor 488 conjugate (Molecular Probes, Invitrogen, Waltham, MA, USA; L21409) was used on the Buin fixed testis tissues. After permeabilization and blocking, the tissues were incubated with PNA-FITC 5 µg/ml for 30 min at room temperature. Mature sperm were fixed with paraformaldehyde (4%) for 20 min at room temperature. After PBS washing, the sperm samples were incubated with 5 µg/ml PNA-FITC and 5 nM Mito Tracker Red (Cell Signaling; 9082) for 30 min at room temperature. The slides were then mounted with DAPI mount. All stainings were performed on minimum of three animals per genotype.

## Transmission electron microscopy

For TEM fresh epididymal sperm and testis tissue were used. After washing with PBS, the samples were fixed in 3% glutaraldehyde at 4°C overnight. The samples were then washed in 0.1 M cacodylate buffer and fixed again in 2% osmium tetroxide at 4% for 2 hr. After dehydration in an ascending ethanol row, the samples were contrasted in 70% ethanol 0.5% uranyl acetate for 1.5 hr at 4°C. Samples were then washed in propylenoxide, three times for 10 min at room temperature before embedding in Epon C at 70°C for 48 hr. Ultra-thin sections were examined using a Verios 460L microscope (FEI) with a STEM III-detector.

## Evolutionary analysis

Evolutionary rates of mammalian Cylicin genes were analyzed according to *Lüke et al., 2016*. Briefly, *Cylc1* and *Cylc2* nucleotide sequences were obtained from NCBI genbank and Ensembl genome browser. Phylogenetic trees of considered species were built according to the 'Tree of Life web project'. The webPRANK software was applied for codon-based alignment of orthologous gene sequences and results were visualized using the ETE toolkit. To determine evolutionary rates of gene sequences across mammals, for different clades and for individual codon sites, the codeml application implemented in the PAML software was used (*Yang, 1997*; *Yang, 2007*). Selective pressures on protein level are represented by calculation of the nonsynonymous/synonymous substitution rate ratio ($\omega$ =dN/dS). It distinguishes between purifying selection ($\omega$ <1), neutral evolution ($\omega$ =1), and positive selection ($\omega$ >1) within various models. The M0 model served as basis for all performed analyses. Different codon frequency settings were tested for the M0 model of each gene and the setting with the highest likelihood was chosen. To test whether alternative models describe the selective constraints within

a dataset more precise than the M0 model, likelihood ratio tests (LRTs) were performed. Applied models and LRTs are described by *Yang, 1997*; *Yang, 2007*; and *Lüke et al., 2016*.

## Study cohort and ethical approval

The MERGE (**M**ale **R**eproductive **G**enomics study) cohort currently comprises over 2030 men, mainly recruited at the Centre of Reproductive Medicine and Andrology (CeRA) in Münster. The large majority has severe spermatogenic failure, that is severe oligozoospermia (<5 Mill./ml sperm concentration), crypto- or azoospermia. So far, only 35 cases were included because of notable sperm morphological defects (≥5 Mill./ml sperm concentration, <4% normal forms). Common causes for infertility such as oncologic diseases, AZF deletions, or chromosomal aberrations were ruled out in advance. Patients with aetiologically still unexplained infertility underwent whole exome sequencing.

All participants gave written informed consent according to the protocols approved by the Ethics Committee of the Medical Faculty Münster (Ref. No. MERGE: 2010-578-f-S) in accordance with the Declaration of Helsinki in 1975.

## Whole exome sequencing and data analysis

After DNA extraction from patients' peripheral blood lymphocytes, WES was performed as previously described (*Wyrwoll et al., 2020*). WES data obtained from 2066 infertile men was filtered for rare (≤0.01 minor allele frequency, gnomAD) variants located within the coding sequence or the adjacent 15 bp of flanking introns in *CYLC1* and *CYLC2*. Patients carrying only one variant in either of the gene were excluded. In case of either bi-allelic *CYLC2* variants or a combination of *CYLC1* and *CYLC2* variants, the whole exome dataset was screened to rule out other potential genetic causes. Variants detected in this study were classified according to the guidelines by the American College of Medical Genetics and Genomics-Association for Molecular Pathology (ACMG-AMP) (*Richards et al., 2015*) adapted to recent recommendations as outlined in *Wyrwoll et al., 2023*.

To rule out an alternative genetic cause for the patient's condition, his exome data was screened for rare (MAF ≤0.01, gnomAD, and in-house database), homozygous, X-linked, or potentially compound heterozygous high-impact variants (stop gain, start lost, stop lost, frameshift, splice site, and splice region as well as missense variants with CADD >20) and rare, heterozygous LoF variants (stop gain, start lost, stop lost, frameshift, splice site) without filtering for a specific set of genes. Respective genes were screened for testis expression and reports of infertility.

## Sanger sequencing

Sanger sequencing with variant-specific primers was conducted for validation and segregation purposes. The primers to confirm the *CYLC1* (NM_021118.3) variant c.1720G>C p.(Glu574Gln) are 5'-ACTGATGCTGACTCTGAACCG-3' (forward) and 5'-CCTTCGAGTCACAAAGTTGTCA-3' (reverse). To confirm the *CYLC2* (NM_001340.5) variant c.551G>A p.(Gly184Asp), the primers 5'-CTGTCGAG GTGGATTCTAAAGC-3' (forward) and 5'-TGCATCCTTCTTTGCATCCT-3' (reverse) were used.

## Analysis of the human sperm samples

Human ejaculate samples from healthy donor and patient M2270 were analyzed according to WHO guidelines prior to washing in buffer and centrifuged (1000 rpm, 20 min). The cells were fixed in methanol and acetic acid (3:1) and used for immunofluorescence staining. Samples were dropped on slides and permeabilized with 0.1% Triton X-100. After blocking with 5% BSA for 30 min, slides were incubated with primary CCIN or CYLC1 antibodies (concentrations shown in *Table 3*) for 3 hr at room temperature. Secondary antibodies were incubated for 1 hr, followed by mounting with DAPI containing medium. All stainings were repeated three times using aliquots of the same sample.

## Statistics

For all analyses mean values ± SD were calculated. Statistical significance was determined by one-way ANOVA using Bonferroni correction. All experiments were conducted as biological replicates and N is provided in Methods section and/or figure legends.

## Acknowledgements

This study was supported by a grant from the German Research Foundation (DFG) to HS (SCHO 503/23-1, project number 458746826), SS (SCHN 1668/1-1, project number: 458746826) and FT (Clinical Research Unit, Male Germ Cells', CRU326). We are grateful to Gaby Beine, Angela Egert, Andrea Jäger, Greta Zech, Luisa Meier, and Christina Burhöi for excellent technical assistance. We would like to thank the Core Facilities for Microscopy and Analytical Proteomics of the Medical Faculty at the University of Bonn for providing support and instrumentation funded by the Deutsche Forschungsgemeinschaft (DFG, German Research Foundation, project numbers: 388169927, 386936527).

## Additional information

### Funding

| Funder | Grant reference number | Author |
|---|---|---|
| Deutsche Forschungsgemeinschaft | SCHO 503/23-1 | Hubert Schorle |
| Deutsche Forschungsgemeinschaft | SCHN 1668/1-1 | Simon Schneider |
| Deutsche Forschungsgemeinschaft | CRU326 | Frank Tüttelmann |

The funders had no role in study design, data collection and interpretation, or the decision to submit the work for publication.

### Author contributions

Simon Schneider, Conceptualization, Funding acquisition, Investigation, Methodology, Writing - original draft, Writing - review and editing; Andjela Kovacevic, Formal analysis, Investigation, Methodology, Writing - original draft, Writing - review and editing; Michelle Mayer, Ann-Kristin Dicke, Formal analysis, Investigation; Lena Arévalo, Resources, Formal analysis, Visualization; Sophie A Koser, Formal analysis; Jan N Hansen, Resources, Software, Visualization; Samuel Young, Methodology; Christoph Brenker, Sabine Kliesch, Gregor Kirfel, Timo Strünker, Resources; Dagmar Wachten, Resources, Software; Frank Tüttelmann, Resources, Funding acquisition, Writing - review and editing; Hubert Schorle, Conceptualization, Resources, Funding acquisition, Writing - original draft, Project administration, Writing - review and editing

### Author ORCIDs

Simon Schneider https://orcid.org/0000-0001-9894-5608
Andjela Kovacevic https://orcid.org/0000-0001-7719-2467
Michelle Mayer https://orcid.org/0000-0001-7576-8514
Ann-Kristin Dicke https://orcid.org/0000-0002-2171-0525
Lena Arévalo https://orcid.org/0000-0002-0681-0392
Christoph Brenker https://orcid.org/0000-0002-4230-2571
Dagmar Wachten https://orcid.org/0000-0003-4800-6332
Timo Strünker https://orcid.org/0000-0003-0812-1547
Frank Tüttelmann https://orcid.org/0000-0003-2745-9965
Hubert Schorle https://orcid.org/0000-0001-8272-0076

### Ethics

All participants gave written informed consent according to the protocols approved by the Ethics Committee of the Medical Faculty Münster (Ref. No. MERGE: 2010-578-f-S) in accordance with the Declaration of Helsinki in 1975.

All animal experiments were conducted according to German law of animal protection and in agreement with the approval of the local institutional animal care committees (Landesamt für Natur, Umwelt und Verbraucherschutz, North Rhine-Westphalia, approval IDs: AZ84-02.04.2013.A429, AZ81-02.04.2018.A369).

Reviewer #1 (Public Review): https://doi.org/10.7554/eLife.86100.3.sa1
Reviewer #2 (Public Review): https://doi.org/10.7554/eLife.86100.3.sa2
Reviewer #3 (Public Review): https://doi.org/10.7554/eLife.86100.3.sa3
Author Response https://doi.org/10.7554/eLife.86100.3.sa4

## Additional files

### Supplementary files
• MDAR checklist

### Data availability
All data generated or analysed during this study are included in the manuscript and supporting files. Custom-made antibodies used in this study were developed against murine CYLC1 and CYLC2. A limited amount of antibody can be provided upon request. SpermQ software used for the analysis of the flagellar beat is publicly available on GitHub (*Hansen et al., 2017*). For the purposes of this study, a new code was written for color-coded figures and is also freely available on GitHub (copy archived at *Hansen, 2023*).

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
