## [Editor Report · eLife assessment]

This study provides **valuable** insights into the role of two under-researched sperm-specific proteins (Cylicin 1 and Cylicin 2). The authors provide **convincing** evidence that they have an essential role in sperm head structure during spermatogenesis, and that their loss leads to subfertility or infertility, with a dose-dependent phenotype. Importantly, the authors identify infertile males with mutations in both Cylicin1 and Cylicin2. Thus, the findings from the mouse models might be applicable to understanding human male infertility with similar structural defects.

---

## [Referee Report · Reviewer #1 (Public Review)]

Mice and humans have two Cylicin genes (X-linked Cylicin 1 and the autosomal Cylicin 2) that encode cytoskeletal proteins. Cylicins are localized in the acrosomal region of round spermatids, yet they resemble a calyx component within the perinuclear theca of mature sperm nuclei. The function of Cylicins during this developmental stage of spermiogenesis (tail formation and head elongation/shaping) was not known. In this study, using CRISPR/Cas genome editing, the authors generated Cylc1-and Cylc2-knockout mouse lines to study the loss-of-function of each Cylicin or all together.

The major strengths of the study are the rigorous and comparative phenotypic analyses of all the combinatorial genotypes from the cross between the two mouse lines (Cylc1-/y, Cylc2-/-, Cylc1-/y Cylc2+/- and Cylc1-/y Cylc2-/-) at the levels of male fertility, cellular, and subcellular levels to support the conclusion of the study. While spermatogenesis appeared undisturbed, with germ cells of all types detected in the testis, low sperm counts in epididymis were observed. Mice were subfertile or infertile in a dose-dependent manner where fewer functional alleles had more severe phenotypes; the loss of Cylc2 was less tolerated than the loss of Cylc1. Thus, loss of Cylc1, and to an even greater extent, loss of Cylc2, leads to sperm structure anomalies and decreased sperm motility. Particularly, the sperm head and sperm head-neck region are affected, with calyx not forming in the absence of Cylicins, the acrosomal region being attached more loosely, and the sperm head itself appearing structurally rounder and shorter. Furthermore, manchette, which disassembles during spermiogenesis, persists in mature sperm of mice missing Cylc2. It is interesting that the study identifies a human male that has mutations in both CYLC1 and CYLC2 genes and suffers from infertility, with similar motility and sperm structure defects compared to the mouse models. CYLC1 in the sperm from the infertile patient sperm is absent, providing evidence that in both rodents and primates, Cylicins are essential for male fertility. Evolutionary analysis of two genes adds an interesting point. The authors show that the reason for the loss of Cylc2 being more severe is due to the higher conservation of Cylc2 compared to Cylc1 in rodents and primates.

Overall, the work highlights the relevance and importance of Cylicins in male infertility and advances our understanding of perinuclear theca formation during spermiogenesis.

---

## [Referee Report · Reviewer #2 (Public Review)]

The work presented in this manuscript focuses on the role of Cylicins in spermiogenesis and the consequences of their absence on infertility. The manuscript is presented in two parts: the first part studies the absence of Cylicins from KO mouse models and shows in mice that both isoforms of Cylicins are necessary for normal spermiogenesis. The evaluation of double heterozygotes is particularly useful for the second part which looks at the presence of mutations in these genes in a cohort of infertile men. A patient with two hemizygous/heterozygous mutations in the CYLC1 and 2 genes, respectively, was identified for the first time and the results obtained with the KO models support the hypothesis of the pathogenicity of the mutations.

In general, the experiments are perfectly performed and the results are clear. Numerous techniques in the state of the art in male reproduction are used to obtain high-quality phenotyping of the mouse models.

The discovery of two concomitant mutations in an infertile patient is very interesting and the work carried out on mice allows supporting that an absence of CYLC1 and a heterozygous mutation of CYLC2 could lead to a phenotype of complete infertility. However, as the mutation on CYLC2 is not identified as pathogenic, the pathogenicity of this mutation remains in question (the authors note this point in the discussion). It would be interesting to see if the mutated amino acid is conserved between different species. In mice, the authors have shown the importance of these proteins on the morphology of the acrosome. What about in humans?

---

## [Referee Report · Reviewer #3 (Public Review)]

The authors tried to study the role of the cylicin gene in sperm formation and male fertility. They used the Crispr/cas 9 to knockout two mouse cylicin genes, cylicin 1 and cylicin 2. They used comprehensive methods to phenotype the mouse models and discovered that the two genes, particularly cylicin 2 are essential for sperm calyx formation. They further compared the evolution of the two genes. Finally, they identified mutations of the genes in a patient. The major strengths are the high quality of data presented, and the conclusion is supported by their findings from the animal models and patients. The major weakness is that the study is rather descriptive without molecular mechanism studies, limiting its impact on the field.

---

## [Author Response]

The following is the authors’ response to the original reviews.

We appreciate very much the comments and suggestions on our manuscript "Cylicins are a structural component of the sperm calyx being indispensable for male fertility in mice and human". According to the comments, we performed a series of further experiments, re-worded and re-wrote several paragraphs and re-structured the manuscript according to the reviewers’ comment. We think that the manuscript is now improved and are looking forward to the further evaluations. We provide a point by point response to all comments and have prepared a version.

**Recommendations for the authors:**

**Editor’s comment:**
1. As pointed out by all three reviewers, it is critical to show the specificity of the antibodies used. The authors should clarify how the specificity of antibodies is tested. Western blot analysis to show the absence of the protein in the knockout is essential.

As suggested by all reviewers, we additionally performed Western Blot analysis on cytoskeletal protein fractions to further verify the specificity of generated antibodies and the generation of functional knockout alleles. Results nicely confirm the results of the IF staining, however, both anti-bodies detected the bands lower than the predicted molecular weight. In addition, Mass Spectrometry was performed to search for the presence of peptides in the cytoskeletal protein fractions. The paragraph reads now as follows:

Line 127-134: Additionally, Western Blot analyses confirmed the absence of CYLC1 and CYLC2 in cytoskeletal protein fractions of the respective knockout (Fig. 1 G), thereby demonstrating (i) specificity of the antibodies and (ii) validating the gene knockout. Of note, the CYLC1 antibody detects a double band at 40-45 KDa. This is smaller than the predicted size of 74 KDa as, but both bands were absent in Cylc1-/y. Similarly, the CYLC2 Antibody detected a double band at 38-40 KDa instead of 66 KDa. Again, both bands were absent in Cylc2-/-. Next, Mass spectrometry analysis of cytoskeletal protein fraction of mature spermatozoa was performed detecting both proteins in WT but not in the respective knockout samples (Figure 1 – supplement 5; Figure 1 – supplement 6).

Specificity of antibodies was additionally proven by immunohistochemical staining, showing a specific staining only in testis sections but not in any other organ tested. The section reads now as follows:

Line 115-117: Specificity of antibodies was proven by immunohistochemical stainings (IHC), showing a specific signal in testis sections only, but not in any other organ tested (Figure 1 – supplement 2)

2. Re-structuring/streamlining of the figures is recommended. Please consider the flow suggested by reviewer #2 and shorten the evolutionary analysis which takes up more space than it adds to the value of the story.

We thank the reviewers and editor for the valuable suggestion. We re-structured the figures as suggested and rewrote the results section accordingly. The evolutionary analysis was significantly shortened.

3. Provide statistics for the imaging analysis such as TEM as only a single representative image is shown.

We agree that the observed morphological defects require a detailed statistical evaluation. TEM analysis was performed to confirm the results from optical microscopy and representative images with high magnification are shown for a detailed visualization of the defects. For additional quantification, we included statistics for IF stainings against calyx proteins CCIN and CapZa (Fig. 2 I-J). For TEM, we added additional images to the supplement (Figure 3 – supplement 1). Furthermore, we quantified the manchette length of step 10-13 spermatids to prove the increased elongation of the manchette in Cylc2-/- and Cylc1-/y Cylc2-/- spermatids (Fig. 5 A-B).

4. Please consider other points raised by the reviewers below to improve the manuscript and provide responses on how the authors have addressed them.

We thank all reviewers for the detailed review of our manuscript and their valuable suggestions, which helped a lot to improve the manuscript. We considered all points raised by the reviewers to the best of our knowledge and hope that the reviewers will approve the manuscript ready for publication. We added a point-by-point discussion of all comments/suggestions hereafter.

**Reviewer #1 (Recommendations For The Authors):**
Major comments:(1) Antibody specificity: Fig 1E - there are some unspecific binding in Cylc2-/- for CYLC2 and in Cylc1/y Cylc2+/- for CYLC1 in the testis (and elongating spermatids in Figure 1 – Supplement 4). Could authors elaborate/comment on this? Western blot analysis would be also helpful to further support the antibody specificity.

The very weak unspecific staining in the testis for CYLC2 (in Cylc2-/-) and CYLC1 (in Cylc1-/y Cylc2+/-) is only present in the lumen of the seminiferous tubules and/or the residual bodies of the testicular sperm cells and can be referred to as background signal. Importantly, the signal is entirely lost in the PT region, proving specificity of the generated antibodies. We added the following paragraph to the results section:

Line 124-127: The generated antibodies did not stain testicular tissue and mature sperm of Cylc1- and Cylc2-deficient males, except for a very weak unspecific background staining in the lumen of seminiferous tubules and the residual bodies of testicular sperm (Fig. 1 F).

Specificity of antibodies was additionally proven by immunohistochemical staining, showing a specific staining only in testis sections but not in any other organ tested.

Line 115-117: Specificity of antibodies was proven by immunohistochemical stainings, showing a specific staining in testis sections only, but not in any other organ tested (Figure 1 – supplement 2)

To further verify the specificity of generated antibodies and the generation of functional knockout alleles, we additionally performed Western Blot analysis on cytoskeletal protein fractions, confirming the results of the IF staining. No unspecific bands were detected in the Western Blot, further supporting the notion that the weak unspecific signals in IF resemble staining artifacts.

The paragraph reads now as follows:

Line 127-132: Additionally, Western Blot analyses confirmed the absence of CYLC1 and CYLC2 in cytoskeletal protein fractions of the respective knockout (Fig. 1 G), thereby demonstrating (i) specificity of the antibodies and (ii) validating the gene knockout. Of note, the CYLC1 antibody detects a double band at 40-45 KDa. This is smaller than the predicted size of 74 KDa as, but both bands were absent in Cylc1-/y. Similarly, the CYLC2 Antibody detected a double band at 38-40 KDa instead of 66 KDa. Again, both bands were absent in Cylc2-/-.

(2) Please provide more interpretation of the gene dosage effect of Cylicin 2. It is not common to see a gene dosage effect in the sperm phenotype as transcripts and proteins can be shared between haploids due to syncytium formation during spermatogenesis.

We agree and we apologize for the misinterpretation. In Cylc2+/- mice expression of Cylc2 was reduced by half but there was no altered phenotype observed. The sentence now reads as follows:

Line 112: In Cylc2+/- animals expression of Cylc2 was reduced by 50 %.

(3) Line 194-196 - the authors say that the sperm are smaller, with shorter hooks and increased circularity of the nuclei, and reduced elongation. Are these statistically significant? There seems to be a high variation in the graph in S2D and the statistical analysis is not given.

We agree, performed statistical analyses, and highlighted significantly altered values for sperm head elongation and circularity in Figure 2 – Supplement 3.

(4) Line 153-164 It is interesting that the absence of Cylc2 affected many parts of sperm structure. I think some ratios of sperm always have a morphological defect in diverse ways, so it is hard to confirm the finding only with a single sperm image. I think that it will be important to do some statistical analysis or at the minimum show more TEM images from TEM.

We agree that the observed morphological defects require a detailed statistical evaluation. TEM analysis was performed to confirm the results from optical microscopy and representative images with high magnification are shown for a detailed visualization of the defects. For additional quantification, we included statistics for IF stainings against calyx proteins CCIN and CapZa (Fig. 2 I-J). For TEM, we added additional images to the supplement (Figure 3 – Supplement 1).

(5) Line 236-242 - I believe that the phenotype described applies to the sperm from Cylc2-/- and Cylc1/y Cylc2-/- animals; however, I think that the Cylc1-/y Cylc2+/- has a more subtle, intermediate phenotype between the WT and the genotypes missing both Cylc-/- alleles.

We agree and we added a quantification of manchette length at step 10-13 to visualize the differences between the genotypes. The section reads now as follows:Line 268-272: Manchette length was measured starting from step 10 until step 13 spermatids and the mean was obtained, showing that the average manchette length was 76-80 nm in wildtype, Cylc1-/Y and Cylc2+/- while for Cylc2-/- and Cylc1-/Y Cylc2-/- spermatids mean manchette length reached 100 nm (Fig. 5 B). Cylc1-/Y Cylc2+/- spermatids displayed an intermediate phenotype with a mean manchette length of 86 nm.

(6) Since CYLC1 staining is absent in Fig 5B, does that mean that the mutation resulted in protein degradation/instability? Is RNA present? Additional biochemical studies of Cyclins demonstrating the deleterious nature of the mutations would strengthen the molecular pathogenesis of the human mutations.

Thank you for raising these important questions. The CYLC1 variant c.1720G>C is predicted to cause the amino acid substitution p.(Glu574Gln). It is, thus, conceivable that the RNA is present but either the protein is degraded or misfolded and, therefore, not detectable by IF. Unfortunately, for personal reasons of the patient, it is currently not possible to receive additional semen samples, preventing additional analyses of the semen, e.g. analysis of Cylicin transcript level.

(7) Strongly suggest shortening the evolutionary analysis - all the corresponding materials are in supplemental while texts are extensive- or even consider entirely omitting. It does not add a lot to the current study.

We agree that the evolutionary analysis was very detailed. However, we think that the results are important to understand the role of Cylicins for male reproduction in general. The results obtained from the mouse model might be transferable to other species, including humans. Further, the results present a possible explanation for the subfertility of Cylc1-deficient mice, in contrast to infertility of Cylc2-deficient males. We shortened the section, the paragraph reads as follows:

Line 287-302: To address why Cylc2 deficiency causes more severe phenotypic alterations than Cylc1deficiency in mice, we performed evolutionary analysis of both genes. Analysis of the selective constrains on Cylc1 and Cylc2 across rodents and primates revealed that both genes’ coding sequences are conserved in general, although conservation is weaker in Cylc1 trending towards a more relaxed constraint (Fig. 6). A model allowing for separate calculation of the evolutionary rate for primates and rodents, did not detect a significant difference between the clades, neither for Cylc1 nor for Cylc2, indicating that the sequences are equally conserved in both clades.

To analyze the selective pressure across the coding sequence in more detail, we calculated the evolutionary rates for each codon site across the whole tree. According to the analysis, 34% of codon sites were conserved, 51% under relaxed selective constraint, and 15% positively selected. For Cylc2, 47% of codon sites conserved, 44% under neutral/relaxed constraint, and 9% positively selected. Interestingly, codon sites encoding lysine residues, which are proposed to be of functional importance for Cylicins, are mostly conserved. For Cylc1, 17% of lysine residues are significantly conserved and 35% of significantly conserved codons encode for lysine. For Cylc2, this pattern is even more pronounced with 27.9% of lysine codons being significantly conserved and 24.3% of all conserved sites encoding for lysine (Fig. 6).

Minor comments:(1) Line 114, 115, 118 à Figure 1D is already well-described in the previous paragraph and thus redundant. Ref Fig 1D, E; but only figure E shows IF. Maybe supposed to be E and F or just 1E?

We apologize for the mix-up with the subfigures. The mentioned paragraph refers to Fig. 1 E-F, which was corrected accordingly.

Line 117-123: Immunofluorescence staining of wildtype testicular tissue showed presence of both, CYLC1 and CYLC2 from the round spermatid stage onward (Fig. 1 E). The signal was first detectable in the subacrosomal region as a cap-like structure, lining the developing acrosome (Fig. 1 E-F, Figure 1 – supplement 3). As the spermatids elongate, CYLC1 and CYLC2 move across the PT towards the caudal part of the cell (Figure 1 – supplement 4). At later steps of spermiogenesis, the localization in the subacrosomal part of the PT faded, while it intensified in the postacrosomal calyx region (Fig. 1 E-F).

(2) Figure 1F - Arguably, IF images show expression of both CYLC1 and CYLC2 to reach/include the acrosome/hook portion of the sperm head, but the diagram does not reflect that. Why is that?

We agree and apologize for the inconsistency. The illustration was adjusted according to the experimental data showing localization of Cylicins in the whole ventral part of the sperm.

(3) Line 124 - PAS staining mentioned on line 124, is not explained (Periodic acid Schiff staining) until line 605

We agree and introduced the abbreviation accordingly. The PAS staining was moved to Fig. 4. The paragraph reads now as follows:

Line 220-222: To study the origin of observed structural sperm defects, spermiogenesis of Cylicin deficient males was analyzed in detail. PNA lectin staining and Periodic Acid Schiff (PAS) staining of testicular tissue sections were performed to investigate acrosome biogenesis.

(4) Some figures are hard to read due to being very small (S1B, 3F).

We agree and we increased the figure size. For former Figure 3F (now figure 4A), insets with higher magnification of representative sperm were added. Insets are additionally shown in Figure 4 – Supplement 1 in higher resolution.

(5) Line 139 Please specify whether the sperm was capacitated or not.

Analysis of the flagellar beat was performed with non-capacitated sperm. We clarified this in the main text:

Line 203: The SpermQ software was used to analyze the flagellar beat of non-capacitated Cylc2-/- sperm in detail 22.

As described in the Material and Methods section, sperm were only activated in TYH medium, prior to analysis:

Line 732-733: Sperm samples were diluted in TYH buffer shortly before insertion of the suspension into the observation chamber.

(6) Line 142-145; The sentence is interrupted strangely, perhaps the authors meant to write:"Interestingly, we observed that the flagellar beat of Cylc2-/- sperm cells was similar to wildtype cells, however, with interruptions during which midpiece and initial principal piece appeared stiff whereas high-frequency beating occurs at the flagellar tip"

We corrected the sentence accordingly.

Line 206-208: Interestingly, we observed that the flagellar beat of Cylc2-/- sperm cells was similar to wildtype cells, however, with interruptions during which midpiece and initial principal piece appeared stiff whereas high frequency beating occurs at the flagellar tip (Fig. 3 C, Video 1, Video 2).

(7) Line 142 -Wrong Figure number. Figure S4A is a phylogenic analysis.

We regret the mix up and corrected the Figure reference accordingly.Line 204-205: Cylc2-/- sperm showed stiffness in the neck and a reduced amplitude of the initial flagellar beat, as well as reduced average curvature of the flagellum during a single beat (Figure 3 – supplement 2).

(8) L146-147 Better placed in Discussion.

We agree, and we omitted this sentence from the results part.

(9) Line 154-156 - The white arrowheads are present in both WT and KO sperm, thus it appears they denote the basal plate, not necessarily the dislocation/parallel position as the current text seems to suggest. Furthermore, the position of the WT and KO sperm is somewhat different with the tail coiling differently, so it is hard to see whether the two are comparable.

We agree and we removed the white arrowhead in the WT sperm picture, and it now depicts only the dislocation of the basal plate in the Cylc2-/- sperm. Due to the morphological anomalies of Cylc2-/- sperm cells, it’s difficult to determine the exact angle of the depicted cell. However, we added more TEM pictures of the sperm cells (3 for WT and 6 for Cylc2-/-) in Figure 3 – Supplement 1.

(10) Line 164 Please describe in detail what mitochondrial damage the readers expect to see from the TEM image.

We evaluated the observed mitochondrial damage in more detail. Unfortunately, the defects described initially seem to be an artifact of apoptotic sperm cells and could not be identified in vital sperm cells in either of the knockout mouse models. We apologize for this misinterpretation, and we deleted this section in the manuscript.

(12) Figure S2A - no WT comparison, difficult to compare without it (mitochondria in Cylc2-/-)

See (10). We evaluated the observed mitochondrial damage in more detail and in comparison to WT. Unfortunately, the defects described initially seem to be an artifact of apoptotic sperm cells and could not be identified in vital sperm cells in either of the knockout mouse models. We apologize for this misinterpretation and we deleted this section in the manuscript.

(13) Line 172-173 - Fig 3C denotes quantification of abnormal acrosome only, however, the text mentions sperm coiled tail being quantified within this graph - which is it? Is it both of them? Or only one of them?

Figure 3 C (now Figure 2G) showed the percentage of abnormal sperm in general comprising acrosomal as well as flagellar defects. We modified the figure and evaluated acrosomal defects and tail defects separately. The results section was changed accordingly and reads now as follows:

Line 152-159: Loss of Cylc1 alone caused malformations of the acrosome in around 38% of maturesperm, while their flagellum appeared unaltered and properly connected to the head. Cylc2+/- malesshowed normal sperm tail morphology with around 30% of acrosome malformations. Cylc2-/- maturesperm cells displayed morphological alterations of head and mid-piece (Fig. 2 F-G). 76% of Cylc2-/-sperm cells showed acrosome malformations, bending of the neck region, and/or coiling of theflagellum, occasionally resulting in its wrapping around the sperm head in 80% of sperm (Fig. 2 F).While 70% of Cylc1-/Y Cylc2+/- sperm showed these morphological alterations, around 92% of Cylc1-/YCylc2-/- sperm presented with coiled tail and abnormal acrosome (Fig. 2 F-G).

(14) Fig 3D - CCIN in the text, cylicin in the figure - this should be consistent. Furthermore, since only the head is being shown, is CCIN ever detected in the WT sperm tail?

We apologize for the inconsistency, and we added the abbreviation “CCIN” to the figure. CCIN is solely detectable in the sperm head of wildtype sperm as published previously. Irregular staining patterns showing signals in the tail region are only observed upon Cylicin deficiency.

(15) Line 199-200 - To say that head of Cylc2-deficient sperm appears less concave seems redundant, likely the observed increased circularity is contributed to by sperm head being less concave in this region; unless there is an extra point that the authors are trying to make and if so, this needs to be elaborated on

We agree and we deleted the sentence from the manuscript.

(16) Figure legend of Fig S3 is wrong. Only S3A and S3B are present, and in the figure legend S3C corresponds to figure S3B.

We agree and corrected the Figure legends accordingly. Due to the re-structuring of the manuscript, Figures and Supplementary figures were re-ordered as well.

(17) Figure 4B - figure legend and/or text should specify that lectin is green and HOOK1 is in red

We specified the figure legend as well as the main text accordingly:Line: 279-281: Co-staining of the spermatids with antibodies against PNA lectin (green) and HOOK1 (red) revealed that abnormal manchette elongation and acrosome anomalies simultaneously occurred in elongating spermatids of Cylc2-/- male mice (Fig. 5 C).

Line: 560-562: Co-staining of the manchette with HOOK1 (red) and acrosome with PNA-lectin (green) is shown in round, elongating and elongated spermatids of WT (upper panel) and Cylc2-/- mice (lower panel).

(18) Line 261-263 - It is difficult to see what is going on with microtubules in these images, as the resolution is low

We increased the pictures and improved their quality. Microtubules are also depicted with letter ‘m’

(19) Line 265-266 - It seems that there is a persistence of manchette, rather than elongation. From these images, I cannot see gaps, and I am not sure where to look for them. This needs to be labelled further and higher-resolution images could be included for clarity.

We agree, although we observed both excessive elongation and persistence of the manchette. The average length of the manchette is now shown in figure 5B.

The paragraph now reads as follows:

Line 235-239: Microtubules appeared longer on one side of the nucleus than on the other, displacing the acrosome to the side and creating a gap in the PT (Fig. 4 C). Whereas elongated spermatids at step 14-15 in wildtype sperm already disassembled their manchette and the PT appeared as a unique structure that compactly surrounds nucleus, in Cylc2-/- spermatids, remaining microtubules failed to disassemble.

The gaps in the perinuclear theca are better visible in TEM micrographs and the description is now in the paragraph describing TEM.

(20) Line 269 Please include the information of "White arrowhead".

We added the information accordingly.

Line 240-242: In addition, at step 16, the calyx was absent, and an excess of cytoplasm surrounded the nucleus and flagellum (Fig. 4 C, white arrowhead).

(21) Line 276-280 This should be placed in the Discussion.

We agree, and we deleted this concluding remark from the results section.

(22) Is Cylc1 and/or Cylc2 conserved/expressed amongst species other than rodents and primates?

Yes, Cylc1 and Cylc2 homologs were identified in *C. elegans* for example. We added a schematic to the introduction showing the protein structure of human, mouse and C. elegans CYLC1 and CYLC2 (Figure 1 – supplement 1).

The section reads now as follows:

Line 73-78: In most species, two Cylicin genes, Cylc1 and Cylc2, have been identified (Figure 1-supplement 1). They are characterized by repetitive lysine-lysine-aspartic acid (KKD) and lysine-lysine-glutamic acid (KKE) peptide motifs, resulting in an isoelectric point (IEP) > pH 10 14, 15. Repeating units of up to 41 amino acids in the central part of the molecules stand out by a predicted tendency to form individual short α-helices 14. Mammalian Cylicins exhibit similar protein and domain characteristics, but CYLC2 has a much shorter amino-terminal portion than CYLC1 (Figure 1-supplement 1).

(23) The whole chapter "Cylc2 coding sequence is slightly more conserved among species than Cylc1" references only supplemental figures/tables. I find this unusual.

We agree, and in order to show the results of the evolutionary analysis more clearly, we moved the panel to main Figure 6.

Line 286-302: To address why Cylc2 deficiency causes more severe phenotypic alterations than Cylc1deficiency in mice, we performed evolutionary analysis of both genes. Analysis of the selective constrains on Cylc1 and Cylc2 across rodents and primates revealed that both genes’ coding sequences are conserved in general, although conservation is weaker in Cylc1 trending towards a more relaxed constraint (Fig. 6 A). A model allowing for separate calculation of the evolutionary rate for primates and rodents, did not detect a significant difference between the clades, neither for Cylc1 nor for Cylc2, indicating that the sequences are equally conserved in both clades.

To analyze the selective pressure across the coding sequence in more detail, we calculated the evolutionary rates for each codon site across the whole tree. According to the analysis, 34% of codon sites were conserved, 51% under relaxed selective constraint, and 15% positively selected. For Cylc2, 47% of codon sites conserved, 44% under neutral/relaxed constraint, and 9% positively selected. Interestingly, codon sites encoding lysine residues, which are proposed to be of functional importance for Cylicins, are mostly conserved. For Cylc1, 17% of lysine residues are significantly conserved and 35% of significantly conserved codons encode for lysine. For Cylc2, this pattern is even more pronounced with 27.9% of lysine codons being significantly conserved and 24.3% of all conserved sites encoding for lysine (Fig. 6 B).

(24) Line 332 - CYCL2 should be CYLC2

We corrected the typo accordingly.

(25) Line 340 The ratio of head defects is different from Table 1 (98% here and 99 % in the table). Please check this information.

We apologize for the inconsistency. We checked the raw data and corrected the table accordingly.

(26) Line 344-345 From figure 5C it is difficult to determine whether the sperm are "headless" or whether the heads are attached to the highly coiled tails next to them

We agree and we quantified the percentage of sperm showing abnormal flagella and a headless phenotype. Furthermore, we added an arrowhead to figure 6C to highlight headless sperm. The paragraph reads now as follows:

Line 335-339: Bright field microscopy demonstrated that M2270’s sperm flagella coiled in a similar manner compared to flagella of sperm from Cylicin deficient mice. Quantification revealed 57% of M2270 sperm displaying abnormal flagella compared to 6% in the healthy donor (Fig. 7 D). Interestingly, DAPI staining revealed that 27% of M2270 flagella carry cytoplasmatic bodies without nuclei and could be defined as headless spermatozoa (Fig. 7 C, white arrowhead; Fig. 7 E).

(27) L367-368 I agree with the authors' logic of this sentence. Although, it is better to show the co-localization of proteins using multi-channel immunocytochemistry. As you mentioned on L369 this will make your finding more obvious. If it is available, please include the colocalization images of the proteins.

We performed the multi-channel staining against Cylicin1 and Calicin, as well as Cylicin2 and Calicin on mouse epipidymal sperm and it’s shown in Figure 2 – supplement 4. Unfortunately, we did not manage to obtain stainings of tissue sections since antibodies against Cylicins and Calicin require different sample processing.

The sentence was added in the section describing calyx integrity:

Line 168-169: In epididymal sperm, CCIN co-localizes with both CYLC1 and CYLC2 in the calyx (Figure 2 – supplement 4).

(28) Line 376 Please keep the abbreviation. "Calicin" "CCIN".

We included the abbreviation accordingly.

Line 377-378: CCIN is shown to be necessary for the IAM-PT-NE complex by establishing bidirectional connections with other PT proteins.

(29) Line 377-378 "Based on ~". The authors did not prove the interaction between CCIN and Cylicins in this article. The mislocalization of CCIN might be resulted in the loss of Cylicins, without any "interaction". To reach this conclusion, a more direct result should be provided.

We agree that we overinterpreted this as we and others did not prove the interaction between CCIN and Cylicins so far. We therefore weakened this statement and formulated it as a hypothesis.

Line 377-381: CCIN is shown to be necessary for the IAM-PT-NE complex by establishing bidirectional connections with other PT proteins. Zhang et al. found CYLC1 to be among proteins enriched in PT fraction 7. Based on their speculation that CCIN is the main organizer of the PT, we hypothesize that both CCIN and Cylicins might interact, either directly or in a complex with other proteins, in order to provide the ‘molecular glue’ necessary for the acrosome anchoring.

(30) Line 499 Please specify which is the target of the immunostaining on the Figure legend. (Tubulin à acetylated α-tubulin)

We specified that α-Tubulin was stained. The figure legend reads now as follow:Line 555-557: Immunofluorescence staining of α-Tubulin to visualize manchette structure in squash testis samples of WT, Cylc1-/y, Cylc2+/-, Cylc2-/-, Cylc1 -/y Cylc2+/- and Cylc1-/y Cylc2-/- mice.

(31) Line 502 Please specify which color indicates which target protein (not only cellular structure).

Line 560-562: Co-staining of the manchette with HOOK1 (red) and acrosome with PNA-lectin (green) is shown in round, elongating and elongated spermatids of WT (upper panel) and Cylc2-/- mice (lower panel).

(32) Line 509 Please include scale bar information in the figure legend like Figure 4 (The magnifications of Figure 5 B, C, and D seem different).

We included the scale bar information accordingly (now Figure 6).

Line 575-588: Figure 6: Cylicins are required for human male fertility

(A) Pedigree of patient M2270. His father (M2270_F) is carrier of the heterozygous CYLC2 variant c.551G>A and his mother (M2270_M) carries the X-linked CYLC1 variant c.1720G>C in a heterozygous state. Asterisks (*) indicate the location of the variants in CYLC1 and CYLC2 within the electropherograms.

(B) Immunofluorescence staining of CYLC1 in spermatozoa from healthy donor and patient M2270. In donor’s sperm cells CYLC1 localizes in the calyx, while patient’s sperm cells are completely missing the signal. Scale bar: 5 µm.

(C) Bright field images of the spermatozoa from healthy donor and M2270 show visible head and tail anomalies, coiling of the flagellum as well as headless spermatozoa who carry cytoplasmatic residues without nuclei. Heads were counterstained with DAPI. Scale bar: 5 µm.

(D-E) Quantification of flagellum integrity (D) and headless sperm (E) in the semen of patient M2270 and a helathy donor.

(F-G) Immunofluorescence staining of CCIN (F) and PLCz (G) in sperm cells of patient M2270 and a healthy donor. Nuclei were counterstained with DAPI. Scale bar: 3 µm.

(33) S2A is not clear. Please describe specifically what the left panel and right panel are about to show with a clear indication of what is PM, mitochondria, etc. On the right, in only one cross-section that shows both mitochondria and the 9+2 axoneme, they look both unaltered whereas on the left, there are unpacked, not aligned mitochondria but the tail boundary is not clear to grasp at first sight.

We apologize for the bad quality of the TEM pictures showing the axonemes and the missing labeling. We recorded and included new images showing an intact 9+2 microtubular structure in Cylc2-/-. Furthermore, we added an image for the wildtype control.

(34) S2D: colors of the last three plots of each graph are too close to tell apart

We agree and changed the color scheme for better visualization.

**Reviewer #2 (Recommendations For The Authors):**
However, I find the manuscript a bit messy, and I will propose to reorganize the figures: following figure 1, showing the reproductive phenotype, I would continue with a figure showing themorphology of sperm in optical microscopy and showing the morphological defect of the nucleus (Fig 3B and 3E), followed with one figure focusing on the flagellum, with images obtained with optical and electronic microscopies, allowing to present the abnormalities of the flagellum and finally the impact on sperm motility and flagellum beating (mix of figure 2FG/3A); next, one figure focusing on acrosome. After that, I would present all data concerning spermiogenesis, starting with figure 2C.

We thank the reviewer for the valuable suggestion, which helps a lot to improve the structure and comprehensibility of the manuscript. We re-organized the figures and the results section accordingly.

Major remarks1. Line 111. The specificity of raised Ab is not clear. Please specify if antibodies are specific: what immune-decorates anti-CYLC1: only CYLC1 or CYLC1 and CYLC2. Same question for anti-CYLC2

Both antibodies were raised against specific peptides of the CYLC1 or CYLC2 protein, respectively. The antigen peptides used for immunization are depicted in the Material and Methods section (AESRKSKNDERRKTLKIKFRGK and KDAKKEGKKKGKRESRKKR peptides for CYLC1; KSVGTHKSLASEKTKKEVK and ESGGEKAGSKKEAKDDKKDA for CYLC2). The peptides used for immunization are specific as they do not resemble the highly conserved and repetitive KKD/KKE motives present in both, Cylc1 and Cylc2.

The specificity of raised antibodies was validated by IF staining of wildype and Cylicin-deficient testis sections. The results clearly show, that CYLC1 signal is absent in Cylc1-deficient spermatids and CYLC2 signal being absent in Cylc2 deficient spermatids.

Specificity of antibodies was additionally proven by immunohistochemical stainings, showing a specific staining only in testis sections but not in any other organ tested.

Line 115-117: Specificity of antibodies was proven by immunohistochemical stainings, showing aspecific staining only in testis sections but not in any other organ tested (Figure 1 - supplement 2)

To further verify the specificity of generated antibodies and the generation of functional knockout alleles, we additionally performed Western Blot analysis on cytoskeletal protein fractions, confirming the results of the IF staining.

The paragraph reads now as follows:

Line 127-134: Additionally, Western Blot analyses confirmed the absence of CYLC1 and CYLC2 in cytoskeletal protein fractions of the respective knockout (Fig. 1 G), thereby demonstrating (i) specificity of the antibodies and (ii) validating the gene knockout. Of note, the CYLC1 antibody detects a double band at 40-45 KDa. This is smaller than the predicted size of 74 KDa as, but both bands were absent in Cylc1-/y. Similarly, the CYLC2 Antibody detected a double band at 38-40 KDa instead of 66 KDa. Again, both bands were absent in Cylc2-/-. Next, Mass spectrometry analysis of cytoskeletal protein fraction of mature spermatozoa was performed detecting both proteins in WT but not in the respective knockout samples (Figure 1 – supplement 5; Figure 1 – supplement 6).

2. Line 115 and figure 1D. From the images presented in figure 1D, it is not clear where CYLC1 and CYLC2 are localized in the round and in elongated spermatids. Please make double staining using a second Ab to identify the acrosome such as DPY19L2 (best option) or SP56 and the manchette such as acetylated alpha-tubulin.

We agree, and we added a double staining of CYLC1/CYLC2 and SP56 to the supplement (Figure 1 - supplement 3), showing co-localization of the developing acrosome and Cylicins. Manchette staining was not performed due to antibodies being available in same species as those against Cylicins (anti-rabbit).

Line 117-120: Immunofluorescence staining of wildtype testicular tissue showed presence of both, CYLC1 and CYLC2 from the round spermatid stage onward (Fig. 1 E, Figure 1 – supplement 3). The signal was first detectable in the subacrosomal region as a cap like structure, lining the developing acrosome (Fig. 1 E-F, Figure 1 – supplement 3).

3. Line 118 and figure 1. The drawing showing the localization of Cylicin in mature sperm does not fit with the experimental data. Cylicins are located on the whole ventral face of the sperm.

We agree and apologize for the inconsistency. The illustration was adjusted according to the experimental data showing localization of Cylicins in the whole ventral part of the sperm.

4. Figure 1: Change "expression of Cylicin" to "localization of cylicin" (green)

We changed the legend accordingly.

5. Line 124 and figure 2C. In the figure provided, the PAS staining seems defective. The acrosomes do not seem stained (in pink as expected for a PAS staining). It may be due to the low quality of the pdf file, nevertheless, it is important to provide in supplementary data, an enlargement of the spermatid region showing the staining of the acrosome.

We apologize for the bad quality of the PDF file and the low magnification. We restructured the subfigure showing PAS stained spermatids at different steps of spermiogenesis at higher magnification. According to the initial reviewer’s suggestion, the PAS staining was moved to figure 4. The PAS staining in figure 2 was replaced by HE-stained overview testis sections in Figure 3 – supplement 1 showing intact spermatogenesis in all genotypes.

6. Line 130. Please indicate a reference for the lower limit of 58%. If this lower limit corresponds to human sperm, it should be omitted.

Indeed, the given reference limit of 58% is only valid for human sperm samples. Therefore, we omitted the reference limit. The paragraph reads now as follows:Line 144-146: Eosin-Nigrosin staining revealed that the viability of epididymal sperm from all genotypes was not severely affected (Fig. 2 D, Figure 2 – supplement 2).

7. line 152 Sperm morphology. Before showing the ultrastructure of the sperm, it would be important to show sperm morphology observed by optical microscopy. Therefore, I recommend including figure S2 as a principal figure, with a mix of Figures 3B and 3E.

We thank the reviewer for the suggestion. The results section was re-structured accordingly, first showing results of optical microscopy (Fig. 2), followed by an in-depth ultrastructural investigation of morphological defects and their effects on sperm motility. Brightfield images of epididymal sperm were moved from former Figure S2 to main Figure 2.

8. Line 164. figure S2A, showing that the 9+2 pattern is normal in KO sperm, is not convincing. Enlargement is required to conclude that the axoneme structure is normal; from the pictures, it rather seems that some doublets are missing.

We apologize for the bad quality of the TEM pictures showing the axonemes. We recorded and included new images showing an intact 9+2 microtubular structure.

9. Line 196. I would suggest removing the term "mild globozoospermia". Globozoospermia is rather complete (100% of round sperm heads) or incomplete (<90 % of round sperm heads). The anomalies observed on sperm heads, sperm motility, and the decrease in sperm concentration are rather suggestive of an OAT.

We agree and we omitted the term “mild globozoospermia”. Instead, we added a concluding remark to the section, summarizing the described defects as OAT syndrome. The section reads now as follows:

Line 215-217: Taken together, observed anomalies of sperm heads, impaired sperm motility, and the decrease in epididymal sperm concentration show that Cylc deficiency results in a severe OAT phenotype (Oligo-Astheno-Teratozoospermia-syndrome) described in human.

10. Line 248. It is not clear from the data of figure 4B that "the developing acrosome lost its compact adherence to the nuclear envelope". From this figure, only defective morphologies of the acrosome are observed

We agree and we omitted the sentence. Furthermore, it does not add additional information to the manuscript, since defects in the attachment of the acrosome to the nuclear envelope have been described in detail in Figure 4C.

11. line 260-264. Manchette defects appear at stages 9-10. At this stage, the HTCA is already attached to the nucleus of the spermatid. see for instance figure 2 from Shang Y, Zhu F, Wang L, Ouyang YC, Dong MZ, Liu C, Zhao H, Cui X, Ma D, Zhang Z, Yang X, Guo Y, Liu F, Yuan L, Gao F, Guo X, Sun QY, Cao Y, Li W. Essential role for SUN5 in anchoring sperm head to the tail. Elife. 2017 Sep 25;6:e28199. doi: 10.7554/eLife.28199 . Therefore, the hypothesis that "abnormal attachment of the developing flagellum to the basal plate and consequently flipping of the head and coiling of the tail in mature spermatozoa" is unlikely and I suggest modifying this paragraph. In the HOOK paper, the manchette defects occurred earlier.

We read the suggested literature and we agree to this reviewer’s comment. Manchette defects that we observe appear at later stages and are probably not responsible for the morphological anomalies of the mature sperm. We also re-analyzed all the manchette staining pictures and didn’t find any defects at earlier stages, so we decided to delete the sentence from the manuscript.

12. Line 344. Please indicate a percentage of headless spermatozoa. Many sperm is too vague.

We agree and we quantified the percentage of sperm showing abnormal flagella and a headless phenotype. The paragraph reads now as follows:

Line 335-339: Bright field microscopy demonstrated that M2270’s sperm flagella coiled in a similar manner compared to flagella of sperm from Cylicin deficient mice. Quantification revealed 57% of M2270 sperm displaying abnormal flagella compared to 6% in the healthy donor (Fig. 7 D). Interestingly, DAPI staining revealed that 27% of M2270 flagella carry cytoplasmatic bodies without nuclei and could be defined as headless spermatozoa (Fig. 7 C, white arrowhead; Fig. 7 E).

13. Any data concerning the success of ICSI for this patient?

Yes, the outcome of the ICSI were added to the main text.Line 309-311: The couple underwent one ICSI procedure which resulted in 17 fertilized oocytes out of 18 retrieved. Three cryo-single embryo transfers were performed in spontaneous cycles, but no pregnancy was achieved.

14. Finally, it would be interesting to study the localization of PLCzeta in this model, since its localization in the perinuclear theca has been clearly shown (Escoffier et al, 2015 doi:10.1093/molehr/gau098 )

We thank the reviewer for the valuable suggestion and performed PLCzeta staining on human sperm, clearly showing an irregular PT staining pattern in sperm of patient M2270 compared to healthy control sperm. Of note, staining was not possible in the mouse due to the antibody being reactive only for human samples.

The section reads as follows:

Line 343-349: Testis specific phospholipase C zeta 1 (PLCζ1) is localized in the postacrosomal region of PT in mammalian sperm (Yoon and Fissore, 2007) and has a role in generating calcium (Ca²⁺) oscillations that are necessary for oocyte activation (Yoon, 2008). Staining of healthy donor’s spermatozoa showed a previously described localization of PLCζ1 in the calyx, while sperm from M2270 patient presents signal irregularly through the PT surrounding sperm heads (Fig. 7 G). These results suggest that Cylicin deficiency can cause severe disruption of PT in human sperm as well, causing male infertility.

**Reviewer #3 (Recommendations For The Authors):**
1. Why the Cylc1-/y Cylc2+/- males were infertile? It would be helpful to show the homologue of the two proteins;

To elaborate more on the homology of CYLC1 and CYLC2, we added a more detailed section about the protein and domain structure to the introduction.

Line 73-78: In most species, two Cylicin genes, Cylc1 and Cylc2, have been identified (Figure 1supplement 1). They are characterized by repetitive lysine-lysine-aspartic acid (KKD) and lysine-lysineglutamic acid (KKE) peptide motifs, resulting in an isoelectric point (IEP) > pH 10 14, 15. Repeating units of up to 41 amino acids in the central part of the molecules stand out by a predicted tendency to form individual short α-helices (Hess et al., 1993). Mammalian Cylicins exhibit similar protein and domain characteristics, but CYLC2 has a much shorter amino-terminal portion than CYLC1 (Figure 1supplement 1).

Speculations about the infertility of Cylc1-/y Cylc2+/- males was added to the discussion:

Line 410-413: Interestingly, Cylc1-/Y Cylc2+/- males displayed an “intermediate” phenotype, showing slightly less damaged sperm than Cylc2-/- and Cylc1-/Y Cylc2-/- animals. This further supports our notion, that loss of the less conserved Cylc1 gene might be at least partially compensated by the remaining Cylc2 allele.

2. Western blot is important to show the absence of the two proteins in the mouse models;

To further verify the specificity of generated antibodies and the generation of functional knockout alleles, we additionally performed Western Blot analysis on cytoskeletal protein fractions, confirming the results of the IF staining.

A paragraph was added to the manuscript and reads as follows:

Line 127-134: Additionally, Western Blot analyses confirmed the absence of CYLC1 and CYLC2 in cytoskeletal protein fractions of the respective knockout (Fig. 1 G), thereby demonstrating (i) specificity of the antibodies and (ii) validating the gene knockout. Of note, the CYLC1 antibody detects a double band at 40-45 KDa. This is smaller than the predicted size of 74 KDa as, but both bands were absent in Cylc1-/y. Similarly, the CYLC2 Antibody detected a double band at 38-40 KDa instead of 66 KDa. Again, both bands were absent in Cylc2-/-. Next, Mass spectrometry analysis of cytoskeletal protein fraction of mature spermatozoa was performed detecting both proteins in WT but not in the respective knockout samples (Figure 1 – supplement 5; Figure 1 – supplement 6).

3. On Page 7, line 227 and line 243, was the acetylated α-tubulin or α-tubulin antibody used?

For all stainings α-tubulin antibody was used. We corrected this accordingly.Line 257-259: We used immunofluorescence staining of α-tubulin on squash testis samples containing spermatids at different stages of spermiogenesis to investigate whether the altered head shape, calyx structure, and tail-head connection anomalies originate from possible defects of the manchette structure.

4. Fig. 2S: A cartoon showing the elongation and circularity of nuclei for evaluation is helpful; The TEM images from the control and Cylc1 KO mice are needed;

Cylc1-/Y TEM picture was added in Figure 3A.

5. The discussion should be rewritten. The current version is to repeat the experiments/findings. The authors should discuss more about the potential mechanisms.

We discussed the observed defects of Cylc-deficient animals and discussed this in relation to other published mouse models deficient in Calyx components. Furthermore, we speculated about potential interaction partners of Cylicins and the importance of these protein complexes for male fertility. However, to this point, we think that it is too farfetched to speculate about potential mechanisms without any evidence for Cylc interaction partner or their exact molecular function. This requires further research.